# SurgicalSemiSeg: A Semi-Supervised Framework for Laparoscopic Image Segmentation

**Yuning Zhou**[1,2]                                          YUNIZHOU@STUDENT.UNIMELB.EDU.AU
**Henry Badgery**[4]                                          HENRY.BADGERY@SVHA.ORG.AU
**Matthew Read**[5]                                           MATTHEW.READ@UNIMELB.EDU.AU
**James Bailey**[3]                                           BAILEYJ@UNIMELB.EDU.AU
**Catherine E. Davey**[1,2,6,7]                               CATHERINE.DAVEY@UNIMELB.EDU.AU

[1] *Department of Biomedical Engineering, The University of Melbourne, Australia*

[2] *Graeme Clark Institute for Biomedical Engineering, The University of Melbourne*

[3] *School of Computing and Information Systems, The University of Melbourne, Australia*

[4] *Department of HPB/UGI Surgery, St Vincent's Hospital Melbourne, Australia*

[5] *Department of Surgery, St Vincent's Hospital Melbourne, Australia*

[6] *Melbourne Brain Centre Imaging Unit, Department of Radiology, The University of Melbourne, Australia*

[7] *Australian National Imaging Facility, St Lucia*

**Editors:** Accepted for publication at MIDL 2025

## Abstract

Deep learning applications in surgery are heavily reliant on large-scale datasets with high-quality annotations, which are costly and time-consuming to obtain. Self-supervised learning (SSL) has shown significant potential for reducing reliance on labelled data. This work investigates the use of SSL for semantic segmentation in laparoscopic cholecystectomy (LC) surgery. Through evaluation of existing SSL methods, we find that pixel-level objectives enable the most effective representation learning for laparoscopic imaging, characterised by highly variable and deformable anatomy. Building on this insight, we develop a tailored masked denoising autoencoder with a carefully optimised masking ratio and patch size for semantic segmentation. Our method achieves state-of-the-art performance across three LC datasets. Of note, it significantly improves segmentation accuracy for critical anatomical structures that are under-represented in training datasets. Furthermore, our approach achieves generalisability, with pre-trained representations performing effectively across fine-tuning datasets from different institutions.

**Keywords:** Self-supervised learning, laparoscopic imaging, semantic segmentation

## 1. Introduction

Deep learning-based precise surgical scene interpretation, such as semantic segmentation, is a crucial component of AI-based intraoperative guidance tools designed to enhance surgical safety. The training of deep neural networks (DNNs) for semantic segmentation requires large-scale datasets with meticulous pixel-level annotations, that are costly and labour-intensive to produce. The development of medical image segmentation datasets includes two major challenges: i) *significant variations in the appearance of anatomical structures and surgical instruments*, and ii) *class imbalance in under-represented structures*. These

challenges have impaired the accuracy of surgical image neural networks, limiting the potential for real world clinical application (Tokuyasu et al.; Maqbool et al., 2020; Silva et al., 2022; Yoon et al., 2022).

Recently, self-supervised learning (SSL) approaches have been employed in surgical computer vision applications to leverage high volume unlabelled data to enhance the performance of DNNs, mitigating the challenges of developing sufficiently large annotated datasets. SSL involves training models on carefully designed pretext tasks using unlabelled data. This pre-trained model can then be fine-tuned on downstream tasks, progressively improving the performance compared to simply training a model on labelled datasets (Chen et al., 2020).

While a recent advancement leverages optical flow for contrastive pre-training on unlabeled laparoscopic data and achieves competitive performance when fine-tuned on less than 10% of annotated samples for semantic segmentation (Moens et al., 2024), most existing studies on SSL focus on classifying structures in surgical imaging (Kletz et al., 2019b; Twinanda et al., 2016; Jin et al., 2018; Mishra et al., 2017; Hashimoto et al., 2019; Kitaguchi et al., 2020). The use of SSL for anatomical and instrument segmentation in surgery remains largely unexplored.

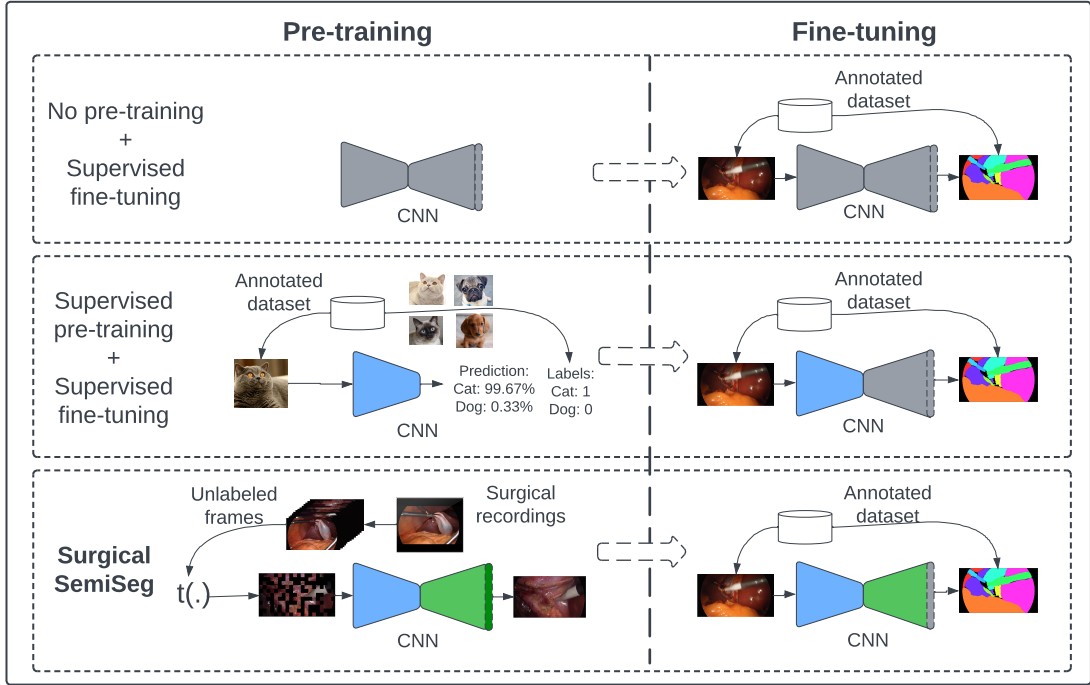

Figure 1: Illustration of two-stage training frameworks for semantic segmentation with three types of pre-training strategies. From top to bottom: no pre-training, supervised pre-training, and our Surgical Semi-supervised Segmentation (SurgicalSemiSeg) framework with tailored denoising autoencoder designs as pre-training. CNN colours indicate the adoption of pre-trained parameters from corresponding architectures for initialising fine-tuning.

In this paper, we evaluate common pretext tasks for static images such as random rotation (Gidaris et al., 2018), colourisation (Zhang et al., 2016), autoencoder (Hinton

and Salakhutdinov, 2006), and denoising autoencoder (Vincent et al., 2008), alongside advanced methods like contrastive learning (SimCLR) (Chen et al., 2020), masked autoencoder (MAE) (He et al., 2022), and a recent contrastive method tailored for LC segmentation (DDA) (Zhou et al., 2024). Through extensive evaluation, we observe that pixel-level generation tasks are effective for segmentation due to their alignment with pixel-level objectives. Building on this observation, we propose our Surgical Semi-supervised Segmentation framework (SurgicalSemiSeg). Figure 1 demonstrates the framekwork schematic. This framework involves a denoising autoencoder for self-supervised pre-training, and a supervised fine-tuning step.

Specifically, we introduce four masking parameters for the denoising autoencoder, each for a distinct input corruption strategy. These parameters provide flexibility in the masking process, allowing it to operate independently of specific token positions as in a masked autoencoder. Additionally, unlike image-level pre-training approaches that disregard the decoder during fine-tuning (Chen et al., 2020), the pixel-level pre-training objective in SurgicalSemiSeg enables full preservation of the pre-trained model. Only the final layer of the decoder is modified to map class predictions during fine-tuning initialisation. This approach ensures that the understanding capability gained by both the encoder and decoder during pre-training is largely retained, maximising the utilisation of unlabelled data to improve segmentation performance in the downstream task.

In summary, the contributions of this paper are as follows:

- We identify that self-supervised objectives at the pixel level are the most effective for segmentation tasks in surgical contexts.

- We propose a masked denoising autoencoder as a pre-training objective to address the unique challenges of surgical imaging. Our analysis shows that varying mask size significantly impacts downstream performance.

- Leveraging the masked denoising autoencoder, we introduce a two-stage Surgical Semi-supervised Segmentation framework (SurgicalSemiSeg). SurgicalSemiSeg outperforms baseline SSL methods across three downstream datasets and significantly improves the recognition of under-represented yet clinically important classes.

## 2. Surgical semi-supervised framework

We propose a masked-corrupted denoising autoencoder as a pre-training objective considering the unique challenges of surgical images in Section 2.1, and then introduce the SurgicalSemiSeg framework in Section 2.2.

### 2.1. Mask-corrupted denoising objective

Surgical image segmentation poses unique challenges: (i) *hard-to-delineate anatomies* (Ferguson et al., 1992; Asbun et al., 1993), (ii) *predominantly reddish contents in the surgical view*, and (iii) *significant perspective and image quality variations due to different operative approaches and settings*. These challenges result in highly similar nearby pixel values within the same frame. Conventional denoising autoencoders apply Gaussian noise to input pixels, with the model inferring corrupted pixels based on surrounding values. However,

the high pixel similarity in surgical images limits representation learning, making it difficult for the model to distinguish object boundaries between anatomies with similar colour characteristics in downstream segmentation tasks.

Inspired by He et al. (2022), which demonstrated exceptional representation learning by reconstructing large missing input patches, we hypothesise that denoising autoencoders can similarly benefit from carefully designed patch-based noise. Corrupting larger image regions rather than scattered pixels increases reconstruction difficulty, requiring the model to infer missing structures or even entire objects from masked-out areas. Furthermore, unlike MAE (He et al., 2022), where masked patch locations are constrained by the grid design in the ViT architecture (Dosovitskiy et al., 2021), denoising autoencoders with CNNs allow flexible placement and sizing of masked patches. This introduces additional challenges to the self-supervised objective, further enhancing representation learning.

This paper proposes a mask-corrupted denoising autoencoder designed explicitly for surgical segmentation. Despite the unique representations and challenges in surgical images, no studies have explored the optimal mask design for improving segmentation performance, particularly for underrepresented yet safety-critical anatomical structures. To address this gap, we introduce four key parameters in the mask design to enable robust and effective representation learning to overcome the challenges of surgical image segmentation.

Given an input image $\boldsymbol{x} \in \mathbb{R}^{w \times h \times 3}$ and a binary mask $\boldsymbol{m} \in \{0,1\}^{w \times h}$, a masked transformation function, $t_{\{\rho,N,s,c\}}$ with four parameters on the input image, is defined as $\boldsymbol{x}' = t(\boldsymbol{x}) = \boldsymbol{x} \odot \boldsymbol{m} + (1 - \boldsymbol{m}) \odot \delta$, where $\odot$ is element-wise multiplication applied to each red-green-blue (RGB) channel, and $\delta$ has matching dimensions with $\boldsymbol{x}$ and contains the replacement value for each masked pixel (default is 0). Four masking parameters are described as follows:

- $\rho \in [0\%, 90\%]$: the ratio of masked pixels, or pixels with 0 values in $\boldsymbol{m}$, among the total pixels in the input. $\rho = 0\%$ simplifies the masked pre-training to an autoencoder.

- $N \in [8, 256]$: the side length of an individual square mask patch or the diameter length of a circle mask.

- $s$: the mask component shape. For simplicity we focus on square and circle masks.

- $c$: the replacing value in $\delta$, also known as mask colour. We adopted black or random colours in the masks for every pixel or mask component.

An illustration of different masking parameters is provoded in Figure 2. $\boldsymbol{x} \in \mathbb{R}^{w \times h \times 3}$ denotes an input image of width $w$ and height $h$ in the RGB space. A DNN model, $f_\theta = h \circ g$, is assumed to be a CNN with an encoder-decoder architecture, parameterized by $\theta$. The encoder, $h$, maps the input, $\boldsymbol{x}$, to a set of deep (or latent) features in the high ($C' \gg 3$) dimensional space, $\boldsymbol{z} = h(\boldsymbol{x})$, where $\boldsymbol{z} \in \mathbb{R}^{w' \times h' \times C'}$, conventionally $w \gg w', h \gg h'$, while $C' \gg C$. The decoder, $g$, generates $\boldsymbol{z}$ to the desired output according to the objective and makes the final predictions through its last layer.

The mask-corrupted input $t(\boldsymbol{x})$ and the clean input $\boldsymbol{x}$ as input-reference pairs are then feed into the CNN training in a self-supervised fashion. The encoder extracts deep representations of the whole $t(\boldsymbol{x})$ with mask corruption as $\boldsymbol{z}' = h(t(\boldsymbol{x}))$. The decoder $g$ then

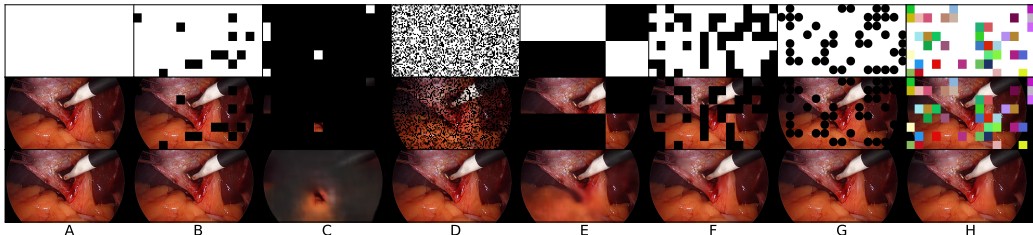

Figure 2: Illustration of mask design parameters on a single example, from top to bottom row, shows the masks, masked images, and reconstructed images under different mask settings ($\rho$ for masking ratio and $N$ for mask size): A. no mask, B. $\rho = 10\%, N = 64$, C. $\rho = 90\%, N = 64$, D. $\rho = 40\%, N = 8$, E. $\rho = 40\%, N = 256$, F. $\rho = 40\%, N = 64$ (optimal mask settings), G. $\rho = 40\%, N = 64$ in circle masks, H. $\rho = 40\%, N = 64$ in coloured masks.

transforms the deep representations of the unmasked input regions (entangled with mask-corrupted noise) back into the input space, as $g(\boldsymbol{z}') \in \mathbb{R}^{w \times h \times 3}$. We adopt the optimisation objective below to minimise the pixel-wise reconstruction differences:

$$\arg\min_{\theta} \mathbb{E}_{\boldsymbol{x} \sim \mathcal{D}_u} \frac{1}{h \times w} \sum_{i=0}^{h-1} \sum_{j=0}^{w-1} \|f_\theta(t(\boldsymbol{x}))_{ij} - \boldsymbol{x}_{ij}\|^2 . \tag{1}$$

Unlike image-level self-supervised objectives, our mask-corrupted denoising objective requires a decoder during pre-training. The decoder learns to capture semantic and spatial relationships between pixels based on the encoder's representation. Pre-training under this objective enables the model to recognise pixels within objects and distinguish them from those between objects within the same image, thereby facilitating accurate pixel-wise classification when fine-tuned on labelled data.

## 2.2. Surgical Semi-supervised Segmentation

We present a simple yet versatile two-stage semi-supervised learning framework named SurgicalSemiSeg, designed to exploit unlabelled surgical images to improve segmentation performance maximally. Figure 1 illustrates the difference between three types of training frameworks for semantic segmentation, including no pre-training, supervised pre-training, and the proposed SurgicalSemiSeg with the tailored mask-corrupted denoising autoencoder as the pre-training objective.

Given a dataset, $\mathcal{D}$ consisting of an unlabelled subset for pre-training, $\mathcal{D}_u$, and a labelled subset for fine-tuning, $\mathcal{D}_l$, we define $\mathcal{D} = \mathcal{D}_u \cup \mathcal{D}_l$, such that $\{\boldsymbol{x}_m\}_{m=1}^p \in \mathcal{D}_u$ and $\{(\boldsymbol{x}_n, \boldsymbol{y}_n)\}_{j=1}^q \in \mathcal{D}_l$. Here, $m$ and $n$ denote the sample sizes of the unlabeled and labeled subsets, respectively, with $m \gg n$ in typical scenarios. An input image is represented as $\boldsymbol{x} \in \mathbb{R}^{w \times h \times 3}$, while its corresponding pixel-wise labels belong to $\boldsymbol{y} \in \mathbb{R}^{w \times h \times K}$, where $K$ is the number of classes.

In the self-supervised pre-training stage, the model is optimised on $\mathcal{D}_u$ following the masked-corrupted denoising objective equation 1.

During supervised fine-tuning, the entire pre-trained model is fine-tuned on $\mathcal{D}_l$. Since the objective shifts from predicting 3-channel RGB values to class probabilities, the output

of the final convolutional layer in the decoder $g$ is adjusted from 3 to $K$, resulting in $g(\boldsymbol{z}') \in \mathbb{R}^{w \times h \times K}$. The fine-tuning process is formulated as the following optimisation problem:

$$\arg\min_{\theta} \mathbb{E}_{\boldsymbol{x} \sim \mathcal{D}_l} \mathcal{L}(f_{\theta}(\boldsymbol{x}), \boldsymbol{y}), \tag{2}$$

where $\mathcal{L}$ is the Cross-Entropy function that measures the discrepancy between predictions and ground truth labels.

The fine-tuning process transfers representations learned during pre-training on large-scale unlabeled surgical video frames to a smaller-scale annotated segmentation dataset. In this study, we focus on static image pretext tasks within this scope. Existing image-level pre-training approaches (Hinton and Salakhutdinov, 2006; Gidaris et al., 2018; Chen et al., 2020) typically re-use only encoder weights during fine-tuning.

SurgicalSemiSeg maximises the retention of self-supervised pre-trained representations by reinitialising only the final layer weights during fine-tuning. This ensures that the semantic and spatial knowledge acquired during pre-training is largely preserved. Equipped with a masked-corrupted denoising autoencoder, SurgicalSemiSeg fully leverages pixel-level self-supervised representation learning for segmentation tasks. With pixel-level objectives in both stages, our framework supports the flexible integration of any segmentation model with an encoder-decoder architecture, making it highly adaptable to various applications. We adopt a CNN in this study due to its flexibility across different input resolutions.

## 3. Experiments

### 3.1. Datasets description

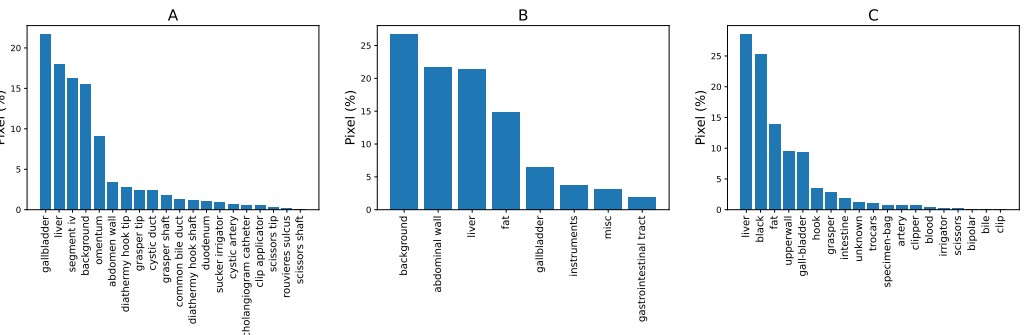

Figure 3: Class distribution of pixels in three LC segmentation datasets, A:In-house Seg B:CholecSeg8k C:M2caiSeg.

We conduct our experiments on 3 public and 2 private datasets. For pre-training, we use public Cholec80 (Twinanda et al., 2016) and private in-house Unlabelled datasets collected from St Vincent's Hospital Melbourne (private and public) and Epworth Healthcare. Our private dataset consists of 300,000 frames at 1920×1080 resolution, collected from 50 LC recordings. For evaluation of the representation quality, we use 3 LC segmentation datasets: an in-house segmentation dataset (in-house Seg) collected from 20 LC recordings from the same hospitals while distinct from those included in in-house Unlabelled, and two public

datasets, CholecSeg8k (Hong et al., 2020) and m2caiSeg (Maqbool et al., 2020). The class distribution statistics of these datasets are shown in Figure 3, with a detailed dataset description provided in Appendix A. All datasets are carefully examined to ensure there is no data leakage between the training set and test set.

## 3.2. Experiment settings

DeepLabV3+ (Chen et al., 2018) with ResNet101 (He et al., 2016) backbone is adopted as the default model. For each pre-training, the model was trained for 20 epochs with 16 (SimCLR with 128) as batch size, AdamW (Loshchilov and Hutter, 2019) as the optimiser, 0.001 as learning rate, and 0.01 weight decay. All fine-tuning processes apply the same parameters, except adjusting the learning rate to 0.005. For computational efficiency, we resized the in-house images to $960 \times 540$ and followed the original resolutions for public datasets. Augmentations of 10 degrees of rotation, horizontal flipping, and colour jittering (with brightness 0.25, contrast 0.25, saturation 0.25, and hue 0.0) were applied. Experiments were conducted on 4 A100 GPUs with PyTorch implementations. Performance is evaluated under class-wise Intersection over Union (IoU) and the mean over all classes (mIoU). Additionally, Dice score is also provided in the appendix. Our implementation is available in this code repository: https://github.com/JoJoNing25/SurgicalSemiSeg.

**Pre-training**: We use the in-house Unlabelled dataset as the default pre-training dataset. A model is pre-trained for each baseline self-supervised strategy on this dataset. Additionally, to explore the optimal masked denoising autoencoder design, we perform a grid-search over different mask parameter settings, leading to 34 pre-trained models. The searching process is illustrated in Figure 4.

**Fine-tuning**: To evaluate the effectiveness of different SSL strategies, we fine-tune the pre-trained models across three LC segmentation datasets. For our mask-corrupted denoising autoencoder, we adopt the optimal mask parameters (the searching process is illustrated in Figure 4, and further described in Section 3.3) searched under In-house datasets as default settings. In-house Seg is split into a training set that contains 3,740 frames from 16 videos, and a representative test set with 392 frames from 4 videos which are carefully selected by surgeons to ensure all classes are presented. For the public benchmarks, we follow the experimental settings of (Silva et al., 2022) and (Maqbool et al., 2020) for class definitions and train-test splits in CholecSeg8k and M2caiSeg.

**Transferability**: To further validate the generalisation ability of SurgicalSemiSeg under its optimal mask settings, we pre-train another model on Cholec80 and evaluate its performance across three segmentation datasets. Results are reported in the last column of Table 1 and the discussion in Section 3.5.

## 3.3. Masked denoising autoencoder designs

To reduce computation cost, we first search for the optimal masking ratio with a fixed $N$ as illustrated in Figure $4(a)$, and then fix the optimal $\rho$ and investigate $N$ as in Figure $4(b)$. For in-house Seg, the optimal performance is reached when masking 40% with fixed $N$ to 64, and the performance is relatively stable between 30% and 60%. Varying $N$ has a more significant influence on downstream performance, which peaked at $N = 64$ when $\rho = 40\%$. We further conducted a grid search with $\rho$ and $N$ in a similar performance

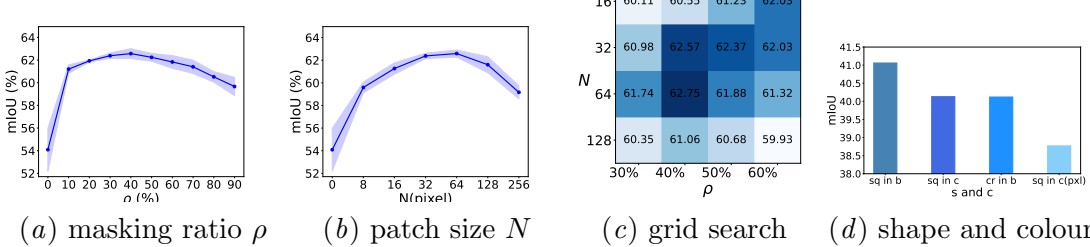

$(a)$ masking ratio $\rho$    $(b)$ patch size $N$    $(c)$ grid search    $(d)$ shape and colour

Figure 4: Influence of varying mask parameters on In-house Seg. Results are reported as mIoU (in percentage). Curves with filling show the mean and standard deviation over 5 random seeds. Darker colour in (c) indicated better performance. Acronyms in (d): $sq$-square mask, $cr$-circle mask, $b$-black mask, $c(pixel)$-random colour on mask pixels

range and confirmed that the optimal setting for in-house Seg is $\rho = 40\%$ and $N = 64$ in Figure 4(c). On this optimal setting, results in Figure 4(d) show the black square mask is preferable. Unless explicitly stated, we use $\rho = 40\%$ and $N = 64$ with the black square mask as the default setting in following sections for In-house Seg based on these observations.

For public datasets, increasing masking ratio and patch size also results in upward parabola in segmentation performance as illustrated in Appendix B. M2caiSeg is less sensitive to ratio changes, but demonstrates higher performance variance across different runnings of the models under the same mask settings, which is a common challenge in deep learning with small dataset. On CholecSeg8k, the optimal $\rho$ is 20% and $N$ at 32.

### 3.4. Comparison with existing pre-text tasks

Table 1: Performance of different pre-training strategies on three validation datasets. $mIoU$ is reported in percentage. RI denoted random initialisation. The best results are in **bold**.

| Fine-tuning datasets | Classes | | Pre-training strategies and datasets | | | | | | | | | |
|---|---|---|---|---|---|---|---|---|---|---|---|---|
| | All | Under-rept. ($<1\%$) | RI | Supervised | Rotation | Colourisation | Autoencoder | SimCLR | MAE | DDA | Ours | Ours |
| | N/A | | N/A | ImageNet | In-house Unlabelled | | | | | | | Cholec80 |
| In-house Seg | 20 | | 56.03 | 59.16 | 56.20 | 59.15 | 50.37 | 58.57 | 61.63 | 58.44 | **62.26** | 60.71 |
| | | 11 | 42.23 | 45.94 | 43.01 | 45.92 | 37.79 | 45.10 | 44.73 | 48.72 | **50.67** | 48.48 |
| CholecSeg8k | 7 | | 57.49 | 61.59 | 55.33 | 64.52 | 56.71 | 57.87 | 61.18 | 61.71 | **66.90** | 64.11 |
| | | 1 | 41.85 | 33.29 | 44.50 | 28.33 | 22.64 | 34.63 | 45.51 | 40.98 | **58.71** | 47.40 |
| M2caiSeg | 19 | | 67.23 | 77.21 | 72.63 | 72.61 | 70.85 | 77.00 | 72.01 | **85.37** | 81.45 | 78.43 |
| | | 12 | 55.30 | 68.16 | 62.32 | 61.81 | 62.26 | 67.51 | 61.74 | **78.60** | 73.61 | 69.69 |

The results, summarised in Table 1, show the average performance of all classes and specifically for under-represented classes, defined as those comprising less than 1% of the pixel distribution (see 3 in Section 3.1). Except for M2caiSeg, which is a very small dataset, pixel-level pretext tasks generally outperform image-level ones. Our method notably improves prediction accuracy, especially for under-represented classes.

We further report the class-wise IoU of different pre-training methods on In-house Seg in Table 2. For critical anatomical structures for safety, such as the common bile duct and omentum, our method improves the baselines by 5.27% and 4.36%. It also significantly improves the recognition of scissors (7.61%), a challenging class easily confused with other instruments (Kletz et al., 2019a; Jaafari et al., 2021; Namazi et al., 2022).

Table 2: Class-wise performance of different pre-training strategies on In-house Seg. In-house Unlabelled is adopted as the default self-supervised dataset. *IoU* and *mIoU* is reported in percentage. RI denoted random initialisation. The best results are in **bold**.

| Categories | Class Names | Pre-training Strategies | | | | | | | | |
|---|---|---|---|---|---|---|---|---|---|---|
| | | RI | Supervised | Rotation | Colourisation | Autoencoder | SimCLR | MAE | DDA | Ours |
| N/A | background | 97.56 | 97.87 | 97.36 | 97.87 | 89.67 | 98.00 | **98.08** | 97.68 | 97.96 |
| Instruments | cholangiogram catheter | 63.76 | 74.80 | 69.02 | 71.16 | 61.24 | 76.36 | 75.86 | 73.50 | **76.97** |
| | clip applicator | 49.96 | **60.73** | 45.39 | 53.78 | 39.10 | 48.68 | 60.52 | 47.72 | 58.01 |
| | diathermy hook shaft | 71.52 | 73.54 | 71.75 | 72.49 | 64.22 | 72.13 | **81.39** | 73.36 | 73.69 |
| | diathermy hook tip | 81.59 | 84.15 | 83.39 | 84.49 | 80.33 | 84.12 | **87.17** | 81.73 | 85.19 |
| | grasper shaft | 76.34 | 78.89 | 76.00 | 79.06 | 64.86 | 79.21 | **80.35** | 78.88 | 80.21 |
| | grasper tip | 57.50 | 64.38 | 54.70 | 64.00 | 53.70 | 65.18 | 64.35 | 61.61 | **67.24** |
| | scissors shaft | 7.60 | 5.93 | 5.20 | 1.54 | 7.70 | 6.06 | **19.11** | 11.66 | 12.27 |
| | scissors tip | 25.36 | 27.67 | 36.11 | 27.75 | 34.03 | 24.33 | **59.47** | 23.91 | 45.27 |
| | sucker irrigator | 55.76 | 63.38 | 52.53 | 62.21 | 45.06 | 62.83 | 64.43 | 63.27 | **65.63** |
| Anatomies | abdomen wall | 36.16 | 41.44 | 34.71 | 41.20 | 12.81 | 39.23 | 39.73 | **42.55** | 41.06 |
| | common bile duct | 55.59 | 50.95 | 57.60 | **59.67** | 45.35 | 56.44 | 59.51 | 56.76 | 56.89 |
| | cystic artery | 27.58 | 25.08 | 27.86 | 32.51 | 24.94 | 28.65 | **33.75** | 25.46 | 33.20 |
| | cystic duct | 49.79 | 50.46 | 48.78 | 50.38 | 46.68 | 50.36 | 51.13 | 51.08 | **52.96** |
| | duodenum | 13.75 | 25.69 | 18.75 | 24.86 | 17.32 | 25.11 | 20.54 | 23.46 | **28.38** |
| | gallbladder | 80.26 | 81.85 | 79.95 | 82.52 | 78.19 | **82.76** | 81.78 | 81.40 | 82.34 |
| | liver | 84.96 | 87.01 | 84.58 | 86.50 | 79.61 | 85.82 | 85.50 | 85.53 | **87.79** |
| | omentum | 87.49 | 87.53 | 87.76 | 88.61 | 72.09 | **88.88** | 88.96 | 88.55 | 88.82 |
| | rouviere's sulcus | 17.26 | 18.70 | 12.90 | 20.10 | 11.95 | 16.35 | 0.00 | 11.83 | **26.85** |
| | segment iv | 80.89 | 83.26 | 79.64 | 82.29 | 78.57 | 80.88 | 81.02 | 83.84 | **84.38** |
| Mean | | 56.03 | 59.16 | 56.20 | 59.15 | 50.37 | 58.57 | 61.63 | 58.44 | **62.26** |

## 3.5. Generalised representations across institutions

The last two columns in Table 1 further demonstrate that our method achieves the best performance when the pre-training and fine-tuning datasets are collected from the same institution, where there is less variation between surgical equipments and operative techniques. Furthermore, the results indicate that representation learning from similar operations, in this case, LC, generalise well across data from different institutions. This finding highlights the potential and effectiveness of leveraging unlabelled surgical recordings to enhance deep learning applications in surgery. While our results specifically validate the method in LC, the approach is likely to perform well across other surgical procedures. Our pre-trained models on Cholec80 is publicly available in this code repository: https://github.com/JoJoNing25/SurgicalSemiSeg.

## 4. Conclusion

In this paper, we conduct an extensive evaluation of self-supervised learning on static image for LC segmentation. Based on our findings that aligned objectives of pre-training and fine-tuning enable the most effective representation learning, we propose SurgicalSemiSeg, a semi-supervised framework with a tailored masked denoising autoencoder for laparoscopic images and provide comprehensive design guidelines. Our method significantly enhances the recognition of under-represented classes that are safety related. This simple yet powerful method offers valuable insights into leveraging unlabelled data for computer-assisted surgery applications. Furthermore, our generalisable and open-sourced pre-trained model serves as a valuable resource for the community, facilitating the development of LC segmentation applications.

## Acknowledgments

This research was supported by The University of Melbourne's Research Computing Services and the Petascale Campus Initiative. All data are provided with ethics approval through St Vincent's Hospital Melbourne and Epworth Healthcare (ref HREC/67934/SVHM-2020-235987).

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

## Appendix A. Dataset description (cont.)

The 5 datasets are comprehensively described in this section, including two unlabelled datasets for pre-training and three labelled datasets for fine-tuning. The collection process are the same with (Zhou et al., 2024)

The two unlabelled datasets adopted for elf-supervised pre-training are described as follows:

- **In-house Unlabelled** contains 300,000 frames at 4 fps from the other 50 videos to avoid data leakage. It is the default unlabelled dataset used for self-supervised pre-training.

- **Cholec80 (Unlabelled)**(Twinanda et al., 2016) is a publicly available classification dataset with tool presence and surgical phase annotations. It contains 80 videos in $480 \times 854$, where each video denotes an individual patient. We only make use of the frames for self-supervised pre-training and disregard the annotations. To avoid data leakage, we also discard the mutual videos in CholecSeg8k and m2caiSeg, which are two subsets of Cholec80 with semantic segmentation annotations. In this way, the adopted unlabeled Cholec80 dataset ends up with 63 videos, which generated 400,000 frames at 2 fps.

The three labelled datasets and their usage are described as below, with the statistics of their class distributions are displayed in Figure 3.

- **In-house Seg** contains frames selected from 20 videos. The individual frames in in 32fps target clips were first pulled out, following by a pixel-wise threshold selection which compares the consecutive frames with the anchor frame and select the next frame if only its pixel difference exceed the threshold. This yielded 4,136 frames in total, where the training set contains 3,740 frames from 16 videos, and the test set contains 392 frames from 4 videos that are unseen in the training set. The dataset was annotated and validated by our surgeons. To evaluate the pre-training strategies and DNNs structure recognition effectiveness in the real-world surgical context, we explicitly defined the semantic class and include *9 surgical instruments* and *10 anatomical structures* shown up during the interested surgical phases. The fine-grained class definition cause the data distribution extremely skewed, which well-represented the real-world challenge.

- **CholecSeg8k** (Hong et al., 2020) is a labeled subset of Cholec80 which contains 8,080 frames of $480 \times 854$ at 25 fps from 17 videos in Cholec80. Following (Silva et al., 2022), we merge the 13 semantic classes into 8 classes under the same train-test split.

- **M2caiSeg** (Maqbool et al., 2020) is a labelled subset of the MICCAI 2016 Surgical Tool Detection dataset (Twinanda et al., 2016), which contains videos from Cholec80. It contains 307 frames in $596 \times 334$ from 2 videos annotated with 19 classes. We follow the same train-test split of the original dataset.

Table 3: Class description and their colour encoding in R, G, B of In-house Seg.

| Class name | Description | Colour code (RGB) |
|---|---|---|
| abdominal wall | abdominal wall | 100, 20, 80 |
| background | black background beyond circular visual field | 200, 200, 200 |
| cholangiogram catheter | instrument to apply dye-enhanced imaging for bile ducts visulization (includes shaft, trip and catheter) | 0, 130, 170 |
| clip applicator | instrument to apply clips to close cystic artery and duct (includes shaft, trip and catheter) | 130, 130, 0 |
| common bile duct | bile duct drain from hepatic ducts to duodenum | 0, 250, 200 |
| cystic artery | blood supply to the gallbladder | 255, 255, 0 |
| cystic duct | duct draining bile from gallbladder to common bile duct | 64, 255, 50 |
| diathermy hook shaft | diathermy hook instrument - shaft | 49,249,166 |
| diathermy hook tip | diathermy hook instrument - tip | 0, 190, 80 |
| duodenum | dection of gastrointestinal tract where common bile duct drains, distal to stomach | 20, 102, 73 |
| gallbladder | gallbladder | 50, 255, 255 |
| grasper shaft | grasping instrument of any kind - shaft | 50, 193, 255 |
| grasper tip | grasping instrument of any kind - tip | 50, 132, 255 |
| liver | all other liver segments | 255, 0, 0 |
| omentum | intra-abdominal fat, includes small bowel | 255, 197, 50 |
| rouviere's sulcus | cleft on the right side of the liver; important landmark | 255, 182, 193 |
| scissors shaft | instrument to cut tissues and structures | 180, 50, 255 |
| scissors tip | instrument to cut tissues and structures | 214, 50, 255 |
| segment iv | segment of liver to the patient left side of gallbladder | 165, 42, 42 |
| sucker irrigator | cylindrical instrument for suction and irrigation | 100, 0, 130 |

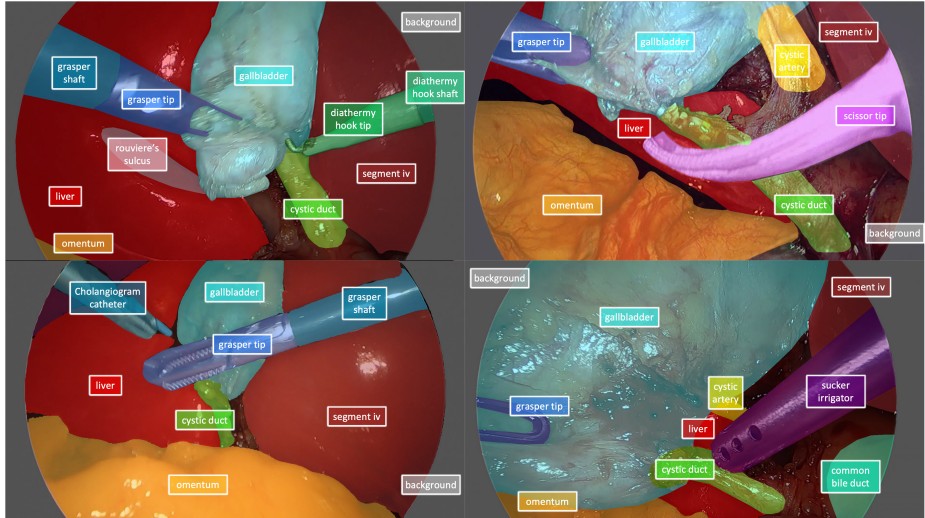

Figure 5: Examples of annotated frames in In-house Seg overlaid with the class colour mask

# Appendix B. Additional results

This section first presents the investigation of the influence of varying the masking ratio and patch size of the denoising masked modelling design on two public datasets, CholecSeg8k and M2caiSeg in Figure 6.

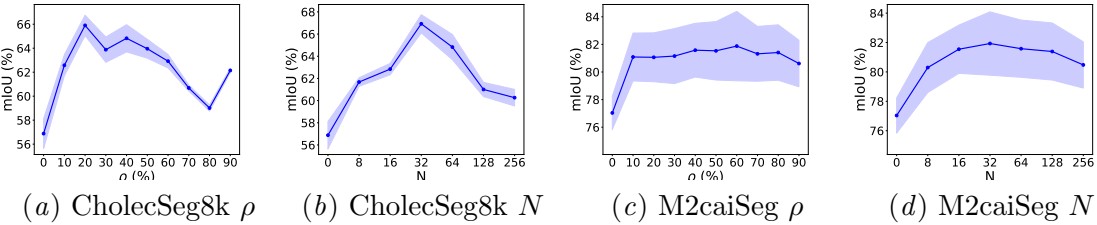

(a) CholecSeg8k $\rho$     (b) CholecSeg8k $N$     (c) M2caiSeg $\rho$     (d) M2caiSeg $N$

Figure 6: Influence of varying mask parameters on CholecSeg8k and M2caiSeg. Results are reported as mIoU (in percentage). Curves with filling show the mean and standard deviation over 5 random seeds. Darker colour in (c) indicated better performance.

It then presents the evaluation results from Table 1 using the Dice score in Table 4 and from Table 2 using the Dice score in Table 5. It also reports the class-wise performance of different pre-training strategies (pre-trained on the in-house Unlabeled dataset) on Cholec-Seg8k in Table 6 and on M2caiSeg in Table 7. Furthermore, it provides a comparison based on the results reported on same classes on CholecSeg8k by Moens et al. (2024) in Table 8.

Table 4: Performance of different pre-training strategies on three validation datasets. In-house Unlabelled is adopted as the default self-supervised dataset. *Dice* is reported in percentage. RI denoted random initialisation. The best results are in **bold**.

| Fine-tuning datasets | Classes | | Pre-training strategies and datasets | | | | | | | | | |
|---|---|---|---|---|---|---|---|---|---|---|---|---|
| | All | Under-repr. ($<1\%$) | RI | Supervised | Rotation | Colourisation | Autoencoder | SimCLR | MAE | DDA | Ours | Ours |
| | | | N/A | ImageNet | | | In-house Unlabelled | | | | | Cholec80 |
| In-house Seg | 20 | | 73.24 | 74.81 | 71.69 | 75.82 | 67.03 | 74.66 | 76.44 | 76.12 | **77.50** | 76.85 |
| | | 11 | 64.94 | 66.38 | 62.68 | 68.19 | 59.23 | 66.72 | 69.70 | 68.51 | **71.61** | 70.22 |
| CholecSeg8k | 8 | | 72.47 | 76.51 | 66.43 | 78.60 | 71.88 | 75.56 | 74.29 | 72.38 | **79.13** | 79.10 |
| | | 1 | 57.54 | 53.28 | 60.35 | 47.40 | 58.87 | 52.30 | 67.82 | 61.16 | **67.88** | 59.69 |
| M2caiSeg | 19 | | 77.87 | 86.97 | 81.17 | 88.97 | 85.34 | 89.33 | 80.13 | **91.67** | 89.91 | 90.47 |
| | | 12 | 68.45 | 80.92 | 72.86 | 84.02 | 78.81 | 84.44 | 71.83 | **87.71** | 85.29 | 86.12 |

Table 5: Class-wise performance of different pre-training strategies on In-house Seg. In-house Unlabelled is adopted as the default self-supervised dataset. *Dice score* is reported in percentage. RI denoted random initialisation. The best results are in **bold**.

| Categories | Class Names | Pre-training Strategies | | | | | | | | |
|---|---|---|---|---|---|---|---|---|---|---|
| | | RI | Supervised | Rotation | Colourisation | Autoencoder | SimCLR | MAE | DDA | Ours |
| N/A | background | 98.76 | 98.92 | 98.66 | 98.92 | 94.76 | 98.99 | **99.02** | 98.79 | 98.97 |
| Instruments | cholangiogram catheter | 79.49 | 85.69 | 81.32 | 84.07 | 77.74 | 86.99 | 86.63 | 84.36 | **87.08** |
| | clip applicator | 66.58 | **78.25** | 64.04 | 70.85 | 57.14 | 66.07 | 77.84 | 65.15 | 77.73 |
| | diathermy hook shaft | 86.11 | 87.47 | 86.31 | 97.03 | 82.49 | 86.32 | **90.44** | 87.55 | 87.58 |
| | diathermy hook tip | 89.85 | 91.37 | 91.15 | 91.57 | 89.32 | 91.38 | **93.15** | 89.90 | 92.06 |
| | grasper shaft | 86.37 | 87.99 | 86.23 | 88.10 | 79.66 | 88.22 | **89.14** | 88.25 | 88.81 |
| | grasper tip | 72.58 | 78.00 | 71.25 | 77.56 | 70.41 | 78.85 | 78.31 | 76.77 | **80.24** |
| | scissors shaft | 49.00 | 23.84 | 20.96 | 43.42 | 28.31 | 41.07 | **70.06** | 46.27 | 42.31 |
| | scissors tip | 61.36 | 58.30 | 58.73 | 59.06 | 59.33 | 48.96 | **76.67** | 58.42 | 65.96 |
| | sucker irrigator | 71.55 | 77.38 | 69.02 | 76.48 | 64.22 | 76.66 | 78.26 | 77.23 | **78.79** |
| Anatomies | abdomen wall | 56.32 | 61.91 | 52.77 | 60.89 | 17.86 | 56.54 | 58.44 | **66.55** | 62.46 |
| | common bile duct | 70.79 | 65.52 | 72.15 | **75.00** | 60.09 | 73.49 | 74.98 | 71.80 | 71.81 |
| | cystic artery | 58.07 | 55.90 | 56.36 | **60.52** | 50.67 | 58.11 | 57.27 | 56.48 | 59.79 |
| | cystic duct | 69.63 | 68.77 | 68.09 | 70.42 | 66.35 | 67.36 | 67.53 | 70.05 | **70.81** |
| | duodenum | 39.70 | 59.14 | 49.88 | 57.97 | 47.28 | 63.82 | 65.45 | 63.60 | **66.30** |
| | gallbladder | 88.82 | 89.82 | 88.65 | 90.25 | 87.63 | **90.45** | 89.81 | 89.62 | 90.14 |
| | liver | 91.92 | 93.07 | 91.68 | 92.76 | 89.09 | 92.44 | 92.28 | 92.14 | **93.51** |
| | omentum | 93.28 | 93.37 | 93.44 | 93.95 | 85.72 | 94.05 | **94.12** | 93.93 | 94.06 |
| | rouviere's sulcus | 45.36 | 50.74 | 44.72 | 47.54 | 44.62 | 44.18 | 0.0 | 54.59 | **50.19** |
| | segment iv | 89.29 | 90.75 | 88.51 | 90.15 | 87.88 | 89.29 | 89.36 | 91.13 | **91.41** |
| Mean | | 73.24 | 74.81 | 71.69 | 75.82 | 67.03 | 74.66 | 76.44 | 76.12 | **77.50** |

Table 6: Class-wise performance of different pre-training strategies on CholecSeg8k. In-house Unlabelled is adopted as the default self-supervised dataset. *IoU* and *mIoU* is reported in percentage. RI denoted random initialisation. The best results are in **bold**.

| Categories | Class Names | Pre-training Strategies | | | | | | | |
|---|---|---|---|---|---|---|---|---|---|
| | | RI | Supervised | Rotation | Colourisation | Autoencoder | SimCLR | MAE | Ours |
| N/A | Background | 97.26 | 97.88 | 97.45 | 97.92 | 97.65 | 97.77 | 97.66 | **97.94** |
| Instruments | Instruments | 59.71 | 60.39 | 41.74 | 69.41 | 49.58 | 59.14 | 57.88 | **69.97** |
| Anatomies | Abdomen Wall | 76.23 | 83.77 | 78.90 | 81.54 | 79.34 | 81.70 | **84.31** | 80.77 |
| | Fat | 81.47 | 86.40 | 81.17 | 83.85 | 85.15 | 79.31 | **88.23** | 87.11 |
| | Gallbladder | 32.60 | 37.36 | 34.47 | **48.51** | 36.14 | 34.38 | 31.81 | 41.26 |
| | Gastrointestinal Tract | 41.85 | 33.29 | 44.50 | 28.33 | 22.64 | 34.63 | 45.51 | **58.71** |
| | Liver | 67.13 | 87.53 | 63.92 | 73.25 | 68.13 | 74.18 | **75.02** | 71.04 |
| | Misc | 3.65 | 17.94 | 0.46 | **33.38** | 15.05 | 1.83 | 9.00 | 28.43 |
| Mean | | 57.49 | 61.59 | 55.33 | 64.52 | 56.71 | 57.87 | 61.18 | **66.90** |

Table 7: Class-wise performance of different pre-training strategies on m2caiSeg. In-house Unlabelled is adopted as the default self-supervised dataset. *IoU* and *mIoU* is reported in percentage. RI denoted random initialisation. The best results are in **bold**.

| Categories | Class Names | Pre-training Strategies | | | | | | | |
|---|---|---|---|---|---|---|---|---|---|
| | | RI | Supervised | Rotation | Colourisation | Autoencoder | SimCLR | MAE | Ours |
| N/A | Black | 96.10 | 96.77 | 96.56 | 95.71 | 97.26 | 96.54 | 97.30 | **97.46** |
| Instruments | Bipolar | 82.22 | **93.08** | 86.19 | 80.50 | 74.47 | 79.98 | 87.87 | 88.56 |
| | Clip | 0.00 | 0.00 | 0.00 | 0.00 | 0.00 | 0.00 | 0.00 | **9.88** |
| | Clipper | 89.66 | 95.98 | 93.64 | 94.68 | 84.50 | 95.31 | 94.31 | **96.34** |
| | Grasper | 82.78 | 89.80 | 83.92 | 88.55 | 84.06 | 88.37 | 82.16 | **90.71** |
| | Hook | 90.52 | 93.84 | 93.66 | 92.80 | 92.60 | 94.81 | 94.36 | **96.30** |
| | Irrigator | 52.88 | 67.34 | 68.47 | 57.35 | 55.64 | 76.82 | 56.99 | **77.30** |
| | Scissors | 0.00 | 0.00 | 0.00 | 0.00 | 0.00 | 2.53 | 0.00 | **2.58** |
| | Specimen-bag | 58.57 | 88.74 | 86.94 | 86.07 | 84.69 | 89.27 | 81.66 | **92.39** |
| | Trocars | 89.97 | 90.37 | 81.64 | 86.20 | 88.72 | 90.00 | 89.03 | **94.32** |
| Anatomies | Artery | 52.28 | 65.08 | 64.03 | 69.65 | 55.71 | 75.37 | 59.71 | **82.93** |
| | Bile | 57.39 | 68.30 | 66.82 | 50 55 | 68.58 | 56.18 | 66.70 | **80.37** |
| | Blood | 53.69 | 76.67 | 71.19 | 73.38 | 66.54 | 75.84 | 73.59 | **87.71** |
| | Fat | 81.89 | 89.06 | 87.89 | 82.72 | 80.81 | 90.40 | 87.79 | **92 95** |
| | Gallbladder | 83.09 | 89.93 | 87.75 | 90.17 | 85.20 | 91.87 | 88.56 | **94.11** |
| | Intestine | 54.40 | **94.23** | 51.73 | 64.64 | 64.40 | 93.62 | 49.05 | 91.24 |
| | Liver | 92.12 | 96.22 | 92.95 | 96.18 | 93.80 | 96.68 | 93.78 | **97.37** |
| | Unknown | 72.63 | 78.07 | 77.23 | 78.72 | 79.86 | 75.23 | **82.06** | 79.69 |
| | Upperwall | 87.17 | 93.55 | 89.32 | 91.66 | 89.24 | 94.15 | 83.27 | **95.44** |
| Mean | | 67.23 | 77.21 | 72.63 | 72.61 | 70.85 | 77.00 | 72.01 | **81.45** |

Table 8: Class-wise performance on 4 classes on CholecSeg8k following 5 baselines reported by Moens et al. (2024). *Dice* is reported in percentage. The best results are in **bold**.

| Method | Fat | Ins | Gb | Bkg |
|---|---|---|---|---|
| LSSL Cholec80 pretrain (16 fine-tune images) (Moens et al., 2024) | 86 | 76 | 62 | 89 |
| Laparoflow-SSL (245 fine-tune images) (Moens et al., 2024) | 84 | 64 | 58 | 86 |
| U-Net++ (Zhou et al., 2018) | 91 | 61 | **63** | - |
| UNETR (Hatamizadeh et al., 2022) | 88 | 71 | 42 | - |
| DeepLabV3+ (Chen et al., 2018) | 86 | 62 | 60 | - |
| Ours (Cholec80 pretrain) | **93** | **86** | 50 | **99** |
| Ours (In-house Unlabelled pretrain) | **93** | 82 | 56 | **99** |

## Appendix C. Dataset granularity

In this section, we explore the effectiveness of our pre-training on the same dataset with different class definitions. Due to the annotation difficulty, it is infeasible for institutions to extensively labelled everything appeared during the surgical procedure. Based on different clinical focus, we defined our extensively labelled dataset with 4 different kinds of granularity of class definitions, with their class distributions illustrated in Figure 7.

- *Only Easy Anatomies* includes 4 anatomies, duodenum, gallbladder, liver, and segment iv. They have relatively larger volume compared to other anatomies and easier to recognse boundaries and appearance.

- *Only Instrument* includes 6 commonly seen instruments in the duration of the procedure we focus, including the cholangiogram catheter, clip applicator, diathermy hook, grasper, sciscissors,nd sucker irrigator. Since it is common in robotic surgery dataset that the surgical tool tips and shafts are annotated separately for precise tool motion tracking, we separate the shaft and tip for three of the most frequently occurring instruments that could possibly cause tissue or organ damage, the diathermy hook, grasper, and scissors.

- *Only Critical Anatomies* includes 5 anatomical structures related mainly to dissection, namely cystic artery, cystic duct, duodenum, gallbladder and Rouvieres sulcus. Within them, the cystic artery and the cystic duct are the two critical structures for dissection. Duodenum is regarded as the danger zone in dissection located below cystic artery, cystic duct. It should be never approached to during the dissection. Rouvieres sulcus is an rarely appeared landmark structure. With its occurrence, the surgical instruments for dissection, like the diathermy hook or scissors, should never operate on any anatomical structures below it.

- *Explicit Classes* consist of 28 classes including the background, surgical instruments and anatomical structures that occurred during the focused operation period. Due to its fine granularity, the class distribution is extremely skewed with 11 classes having fewer than 1% pixels among the entire dataset.

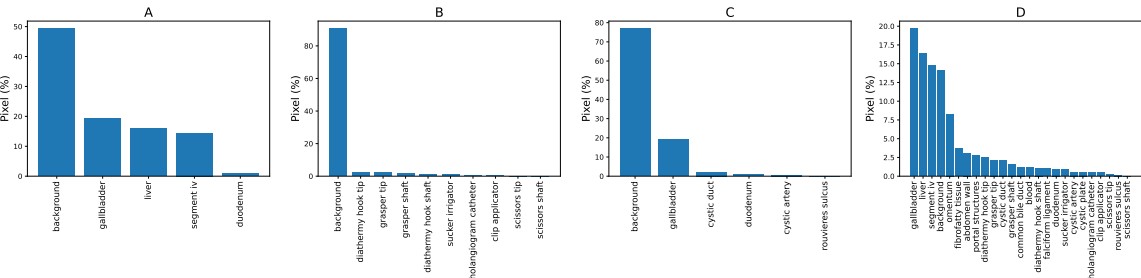

Figure 7: Class distribution of In-house Seg datasets with four kinds of class granularity. A.*Only Easy Anatomies*, B.*Only Instrument*, C.*Only Critical Anatomies*, D. *Explicit Classes*

Table 9: Comparison of pre-training strategies on In-house Seg with different class granularity. *IoU* is reported in percentage. The best results are in **bold**.

| Class definition | Class numbers | Performance |
|---|---|---|
| Only Instruments | 10 | 67.46 |
| Only Critical Anatomies | 6 | 65.07 |
| Only Easy Anatomies | 5 | 71.82 |
| Explicit Classes | 28 | 44.80 |
| Major Classes | 20 | 62.26 |

From Table 9, we observe that performance tends to decline as the number of classes in a dataset increases. Intuitively, defining more classes leads to finer-grained segmentation, which introduces greater semantic complexity, such as smaller structural regions and more rigid boundaries. This suggests that dataset construction should carefully balance annotation cost against the desired level of performance.

## Appendix D. Visualisation on public datasets

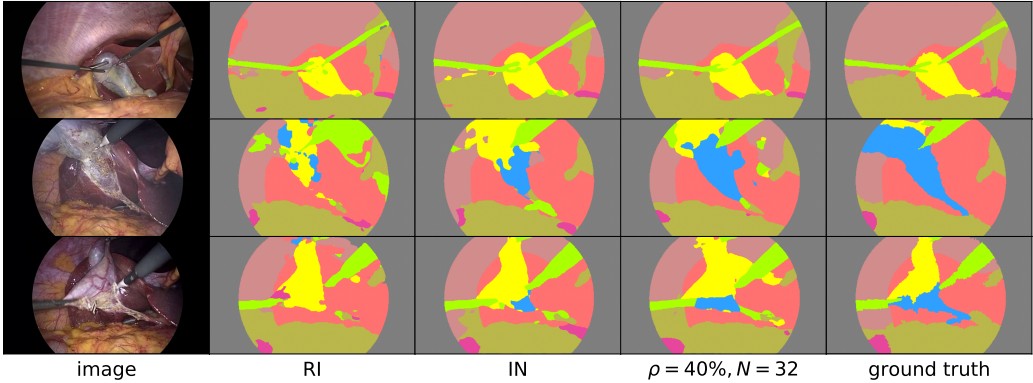

Figure 8: Predictions on three sample images from CholecSeg8k. From left to right shows the original images (image), predictions from no pre-training (RI: random initialisation), supervised pre-training (IN: ImageNet pre-trained), our method ($\rho = 40\%, N = 32$), and the ground truth segmentation masks (ground truth). The colour code follows the original dataset.

Figures 8 and 9 present qualitative comparisons of segmentation performance on the CholecSeg8k and m2caiSeg datasets, respectively. Each row shows predictions from different methods: from left to right, the original laparoscopic image, segmentation results from models trained with random initialisation (RI), supervised pre-training (IN), the proposed method with $\rho = 40\%$ and $N = 32$, and the ground truth mask.

In Figure 8, predictions from RI show noisy and inconsistent boundaries, failing to segment anatomical regions accurately. Supervised pre-training improves structural awareness but still exhibits visible errors in fine-grained details. The proposed method produces cleaner masks with sharper boundaries and greater alignment with the ground truth, especially in complex regions with overlapping instruments and tissues.

In Figure 9, RI again results in coarse, error-prone predictions, while IN shows slightly improved consistency. The proposed method demonstrates clearly improved segmentation, particularly in preserving small structures and achieving better boundary precision. Its outputs closely match the ground truth annotations across all examples.

Overall, these figures show that the proposed semi-supervised pixel-level pre-training method consistently outperforms models trained from scratch or with standard pre-training, particularly in producing reliable and anatomically meaningful segmentations under limited supervision.

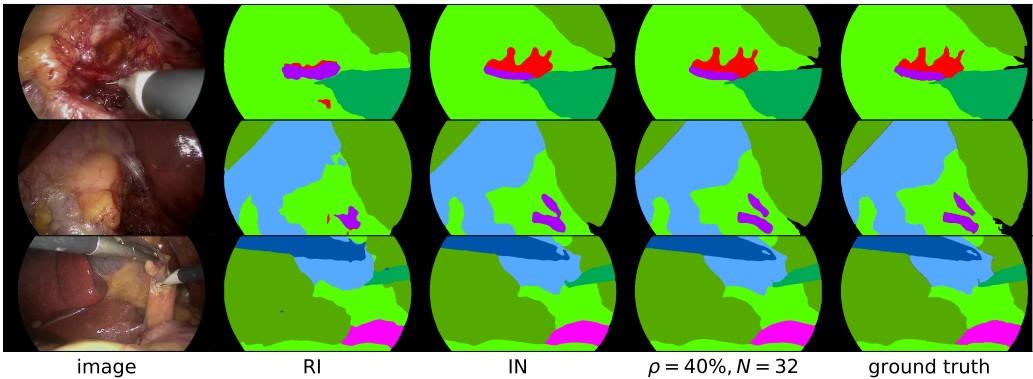

Figure 9: Predictions on three sample images from m2caiSeg. From left to right shows the original images (image), predictions from no pre-training (RI: random initialisation), supervised pre-training (IN: ImageNet pre-trained), our method ($\rho = 40\%, N = 32$), and the ground truth segmentation masks (ground truth). The colour code follows the original dataset.

