# OpenReview forum: "SurgicalSemiSeg: A Semi-Supervised Framework for Laparoscopic Image Segmentation"
_MIDL.io/2025/Conference — MIDL 2025 Poster_

### Official Review · Reviewer_ebsp · 2025-02-15

**Confidence:** 4
**Preliminary Rating:** 3
**Recommendation:** Poster
**Final Rating:** 4

**Summary:**

This paper explores self-supervised learning (SSL) for laparoscopic image segmentation and introduces a masked denoising autoencoder (MDAE) as a pre-training strategy to enhance segmentation accuracy, particularly for underrepresented anatomical structures. The study evaluates SSL objectives across multiple LC datasets, demonstrating improved performance on low-frequency categories and robust cross-dataset generalization. However, the paper does not adequately justify why the proposed SSL is uniquely suited for surgical imaging. Furthermore, the baseline experiments lack transparency, raising concerns about the fairness and reproducibility of the results.

**Strengths:**

1. Exploring SSL in medical image segmentation: The paper investigates SSL pre-training strategies for laparoscopic image segmentation, a promising direction given the high annotation costs and limited labeled data in medical imaging.

2. Comprehensive evaluation across multiple LC datasets: The experiments cover CholecSeg8k, M2caiSeg, and In-house Seg, assessing the generalization ability of SSL models across different institutional datasets.

3. Effectiveness of pixel-level SSL objectives: The study shows that pixel-level SSL tasks (MDAE) outperform image-level contrastive learning (e.g., SimCLR, MAE) in segmentation, particularly for low-frequency anatomical classes, demonstrating the importance of pixel-wise learning signals.

**Weaknesses:**

1. Unclear motivation for SSL in surgical imaging: The paper does not convincingly explain why SSL is particularly advantageous for surgical images, as many of its challenges (e.g., deformation, class imbalance) also exist in general segmentation tasks.

2. Lack of adoption Transformer-based: While CNNs are used as the backbone, most modern SSL pre-training approaches (e.g., MAE, DINO) leverage Transformers (e.g., Swin Transformer, ViT). The lack of adoption leaves open questions about the optimal architecture choice for laparoscopic SSL.

3. Insufficient details on baseline experiment setup: The paper does not specify whether all baseline SSL methods were pre-trained on the same dataset or whether supervised baselines were initialized randomly or from existing pre-trained weights. This lack of clarity affects the fairness and interpretability of the comparisons.

**Detailed Comments:**

1. Clarify the unique benefits of SSL for surgical imaging: The paper should better explain why the proposed SSL is particularly beneficial for laparoscopic images and compare it with supervised or semi-supervised approaches to justify its necessity.

2. Include Transformer-based comparisons: Given that Transformers dominate modern SSL research, a comparison with Swin Transformer or ViT would provide stronger evidence regarding CNN’s suitability for surgical image SSL.

3. Ensure transparency in experimental setup: The authors should clearly state whether all baseline methods use the same pre-training data, and whether supervised baselines were initialized randomly or pre-trained on external datasets, ensuring fairness and reproducibility.

**Justification Of The Final Rating:**

After receiving the detailed rebuttal from the authors, I am satisfied with their responses and have decided to upgrade my rating to weak accept.

The authors have provided comprehensive and convincing answers to my key questions regarding the unique challenges in surgical imaging, the rationale for choosing CNN over Transformer, the confirmation that all baseline methods were pre-trained on the same dataset, and the training settings. Furthermore, the authors committed to adding Transformer-related experiments in the final version, which will make the paper more comprehensive.

While the work still has some limitations, the authors' responses have addressed my main concerns. Considering the special challenges faced in the surgical imaging domain, I believe this work makes a valuable contribution to the field of computer-assisted surgery and medical image analysis.

**Justification Of The Preliminary Rating:**

While this paper presents a meaningful exploration of SSL for laparoscopic segmentation, it lacks a strong justification for why SSL is uniquely suited for surgical images, given that similar challenges exist in general segmentation tasks. Additionally, unclear experimental details regarding baseline pre-training raise concerns about fairness and reproducibility. Improving motivation, experimental fairness, and architectural comparisons would significantly strengthen the paper.

**Questions To Address In The Rebuttal:**

1. What specific challenges in surgical imaging make SSL particularly necessary, rather than general segmentation techniques?
2. Why was a CNN chosen instead of a Transformer? Would architectures like Swin Transformer or ViT perform better?
3. Are all baseline methods pre-trained on the same dataset? Were supervised models trained from scratch or using pre-trained weights?

**Special Issue:**

No

---

> ### Author Response · Authors · 2025-03-08
> **Response to Reviewer ebsp**
>
> Thank you very much for reviewing our paper and the valuable comments. We have prepared a clarification for each of your questions, please kindly let us know if anything is still unclear.
>
> --------------------------------------------------------------------
>
> **Q1:** Specify surgical imaging challenges (besides deformation or class imbalance that also exist in general segmentation tasks) that make SSL particularly necessary
>
> **A1:** Below, we outline some unique challenges presented in surgical imaging and how SSL can help overcome them. We have also updated the corresponding paper sections to reflect these points.
>
> **1. Scarcity of annotated data**
>
> Pixel-level annotations for surgical segmentation are expensive and time-consuming, requiring expert surgeon involvement. It can take 7–20 minutes for a surgeon or resident to annotate a single frame [1], and their valuable time is prioritised for saving lives in the operating room.
>
> SSL enables domain-specific pre-training on unlabelled surgical data, improving generalisation across different hospitals, surgeons, and procedures without requiring extensive labelled datasets.
>
> **2. High variability in surgical data and their interpretations**
>
> Surgical scenes are highly diverse due to diverse patient anatomies, instruments, camera angles, lighting conditions, and surgeons’ operation styles. Even experts may have subjective differences in labeling and interpreting surgical events. These problems pose a challenge for the development of accurate DNN, especially under limited annotations.
>
> SSL adopts vast amounts of unlabelled surgical recordings to improve model generalisation. The model can learn domain-invariant features and become more robust to differences in patient anatomy, surgical style, and video quality. Not limited by annotation cost, the training data are not limited in certain important time stamps selected by human annotators. This can possibly alleviate the impact of subjective annotations.
>
> **3. Lack of standardisation across datasets**
>
> Surgical datasets from different institutions lack standardisation in terms of image quality, annotation protocols, surgical instrument models, and operational setups (e.g. the American vs. French positions), making cross-dataset generalisation difficult.
>
> Pre-trained models on large-scale publicly available or in-house datasets can be open-sourced, allowing researchers to fine-tune them for specific surgical applications. This not only reduces the dependency on large-scale labelled datasets, but has the potential to facilitate cross-institution collaboration without the necessity of data sharing.
>
> --------------------------------------------------------------------
>
> **Q2:** Why was a CNN chosen instead of a Transformer?
>
> **A2:**  As mentioned in **Section 2.2** (revised section), our framework supports flexible integration of any encoder-decoder segmentation model, making it adaptable to different applications. We adopt a CNN encoder in this study due to the following key considerations:
>
> **1. Flexibility in handling high-resolution images**
>
> Laparoscopic images have high resolution and significant scale variations, requiring multi-scale feature extraction for both self-supervised pre-training and segmentation. CNNs can process images at varying resolutions without requiring fixed patch sizes, allowing for different input sizes in pre-training and fine-tuning, optimising both performance and computational efficiency.
>
> In contrast, ViTs require fixed token sizes (e.g., 16×16, 32×32 patches), making them less adaptable to high-resolution images. Smaller tokens increase computational cost, while larger tokens lose fine details, limiting their effectiveness in surgical image segmentation.
>
> **2. Training feasibility & computational efficiency**
>
> Maintaining high input resolution is essential for learning fine anatomical structures in self-supervised learning, but this can create a computational bottleneck. CNNs scale linearly O(n) with input size, making them efficient in high-resolution demands.
>
> Conversely, ViTs scale quadratically O(n²) due to global self-attention, leading to high memory and compute demands.
>
> We will provide the full evaluations with ViT for the finalised version in the paper.

---

> ### Author Response · Authors · 2025-03-08
> **Response to Reviewer ebsp**
>
> **Q3:**  Are all baseline methods pre-trained on the same dataset? (and the reproducibility concern)
>
> **A3:** All methods are pre-trained on the same dataset, for every table presented in the paper. There is no unfair comparison. We have revised the paper (please see **Section 3.1** about the dataset usage) and **Table 1** to make this clearer.
>
> The source-code and pre-trained model on Cholec80 will be made publicly available to help with reproducibility.
>
>
> --------------------------------------------------------------------
>
> **Q4:**  Were supervised models trained from scratch or using pre-trained weights?
>
> **A4:** Random initialisation (denoted as *RI* in Table 1 & Table 2) indicates no pre-training was applied. Supervised pre-training (denoted as *Supervised* in Table 1 & Table 2) indicates the application of image-level supervised pre-training on ImageNet, a large-scale labeled dataset.
>
> We have updated **Table 1** in the paper’s revised version to include the applied pre-trained dataset for different pre-training strategies.
>
> --------------------------------------------------------------------
>
>
> [1] Scheikl, P. M., Laschewski, S., Kisilenko, A., Davitashvili, T., Müller, B., Capek, M., ... & Mathis-Ullrich, F. (2020, September). Deep learning for semantic segmentation of organs and tissues in laparoscopic surgery. In Current Directions in Biomedical Engineering (Vol. 6, No. 1, p. 20200016). De Gruyter

---

### Official Review · Reviewer_EccT · 2025-02-18

**Confidence:** 5
**Preliminary Rating:** 1
**Final Rating:** 1

**Summary:**

This paper uses a masked pretraining strategy for surgical segmentation. From the main text, the authors use an in-house LC dataset and a public LC dataset for pretraining and fine-tune their model on three datasets (one in-house and two public). The authors report that their proposed method achieves the best overall segmentation performance.

**Strengths:**

The paper compares different pretraining strategies for surgical segmentation.

Multiple classes are involved in the experiments, and this is important for video segmentation.

Multiple datasets are used to conduct the experiments.

**Weaknesses:**

There is limited novelty in this work. The authors claim a novel mask design, but I dont see that. Thus, this paper appears to be more of a validation study,which makes implementation details particularly important

There is insufficient dataset information in the main text. The main text should provide enough details for readers to evaluate the results section. However, without such information, it is difficult to assess the findings. For example , the test sample size is unknown, and no standard deviations are reported.

I don't understand why fine-tuning only the last layer would be effective for a segmentation task. Additionally, it is not clear what the "last layer" refers to. a convolutional layer or a activation layer? If it refers to the convolutional layer, does that imply this segmentation task is relatively easy?

Compared to MAE, the proposed method doesn't show a significant performance difference for structures with Dice > 60. This is similar to the "supervised" results reported in the table. The mean Dice score might be biased by classes with low Dice, and I don't think those scores matter

The preliminaries section is unnecessary, and I believe readers for our conference will have such background knowledge. I also don't prefer major revisions during the rebuttal phase, and I am not requesting that. It would be better to put more context related to experiments.

It is unclear how many models were trained during pretraining and fine-tuning. Was a single unified model trained, or were separate models trained for each dataset?

An interesting finding from table that the background Dice is not close enough to 1, I don't get this since the area of background is pretty large.

The discussion of missing class is not clear to support the results.

Below is a minor point but important
The backbone selection may not be the best fit for medical image segmentation. This also applies to other experimental settings, such as the loss function. By optimizing these settings, the small differences between the compared methods could be eliminated. Please consider some widely used methods such as nnUnet.

**Detailed Comments:**

please see weaknesses

**Justification Of The Final Rating:**

My rating remains same after the discussion session, as I felt that it was not a productive discussion for me, even for the second round. Most of my questions were not addressed during this period, except for Q6. Even some simple questions, such as Q4 and Q10, remained unanswered. Specifically, the meaning of "mean" and "overall" in Q4, as well as "model"/"network architecture" and "backbone" in Q10. The network architecture is unclear to me. Based on my understanding, the backbone refers to the encoder. Additionally, the paper only mentions DAE instead of U-Net, which raised my original question

Moreover, the author's statement, "As far as we know, standard practice in medical and natural image segmentation does not use SDs to indicate whether a class was present," conflicts with their claim that "the recognition of these underrepresented classes is a well-recognized challenge in surgical contexts." Since this scenario can be categorized as part of a detection setting, incorrect predictions can lead to an IoU of zero, which would certainly affect the standard deviation.

My biggest concern is Q7, where the author's discussion is weak and does not sufficiently support their argument.

**Justification Of The Preliminary Rating:**

My rating is based on the limited novelty and the unclear description of experiments. Without sufficient details, it is difficult to identify the paper's contributions. The paper needs further refinement. There is no option for "reject" and this paper is blow my bar of "weak reject"

**Questions To Address In The Rebuttal:**

please see weaknesses

---

> ### Author Response · Authors · 2025-03-08
> **Response to Reviewer EccT**
>
> Thank you very much for reviewing our paper and the valuable comments. We have prepared a clarification for each of your questions, please kindly let us know if anything is still unclear.
>
> ---
>
> **Q1:** Clarification of the contribution
>
> **A1:** While no completely new self-supervised learning technique was introduced, our finding on the most suitable pre-training task for LC segmentation and the SurgicalSemiSeg framework is a novel and valuable contribution.
> - We find DAE with patch-based mask design significantly improves the representation learning effectiveness for segmentation.
> - We extensively searched for mask patch size, masking ratio, mask colour and shape, and identified that mask size as the most important factor that impacts downstream segmentation performance.
> - Our framework preserves **entire** pre-trained encoder-decoder representation to fully leverage pixel-level self-supervised representations.
> - Our framework significantly improves the recognition of under-represented yet clinically important classes and the pre-trained representations are highly transferable across datasets from different institutions.
>
> We have updated the main contributions in **Section 1** in the revised submission.
>
> For the implementation details, we further clarify them in the following reply in Q5/A5. We have also revised and updated the corresponding paper sections (**Section 3.2**).
>
> ----
>
> **Q2:** Revised dataset information
>
> **A2:** The detailed dataset description is included in Appendix.A. We re-arranged it to **Section 3.1** in the main paper of the updated submission.
>
> We reported the standard deviations of the searching process for the optimal mask design in SurgicalSemiSeg, pre-trained and fine-tuned on the in-house datasets with five different random seeds, in Figure 4. This has now been adjusted to **Section 3.3** of the updated submission.
>
> ---
>
> **Q3:** Clarification of last layer and why fine-tuning only the last layer would be effective for a segmentation task
>
> **A3:** The “last layer” refers to the final layer that generates the model’s output. In the encoder-decoder architecture used for DAE, this corresponds to the **last convolutional layer**, where the output is modified from predicting pixel colors (RGB) to logit class probabilities by changing the output channels from 3 to the number of classes. Additionally, **we fine-tune the entire model**, and just randomly re-initialise the last layer, which has been stated in Section 3.1. We have made this clearer in our revision in **Section 2.2**.
>
> > does that imply this segmentation task is relatively easy?
>
> We do not suggest that this segmentation task is easy; on the contrary, it remains highly challenging. Our framework, which includes pre-training, reinitialising the last layer, and full fine-tuning, is well-suited for LC segmentation but does not inherently simplify the task.
>
> However, in relative terms, our framework allows more straightforward transitioning of DAE representations to the downstream segmentation objective compared to image-level pre-training methods. Take contrastive learning, transitioning the objective from predicting three pixel values to class probabilities is inherently simpler than contrastive learning, which discards the projector and attaches an entirely new segmentation head on top of the backbone.
>
> ---

---

> > ### Comment · Reviewer_EccT · 2025-03-11
> >
> > O denotes original comments
> >
> > 1. O: There is limited novelty in this work. The authors claim a novel mask design, but I dont see that. Thus, this paper appears to be more of a validation study,which makes implementation details particularly important
> >
> > regarding to "We propose a novel mask design with four parameters to enable the application of patch-based masks with varying sizes and masking ratios across the entire image"
> >
> > 2. Including the necessary dataset information in the appendix may be unfair to other authors, as it could potentially influence the review process by requiring additional pages. Additionally, reviewers can choose to make their judgments without referring to the appendix.
> >
> > 3. O: I don't understand why fine-tuning only the last layer would be effective for a segmentation task. Additionally, it is not clear what the "last layer" refers to. a convolutional layer or a activation layer? If it refers to the convolutional layer, does that imply this segmentation task is relatively easy?
> > Re: "Existing image-level pre-training usually re-uses only encoder weights in fine-tuning. Our framework benefits from the semantic understanding and spatial reconstruction ability from the pre- trained decoder by modifying its last layer only."
> >
> > In standard fine-tuning, the final layer is often reset when transferring to a new dataset or task, which I had initially pointed out in my comments. That’s why I found the explanation confusing. Additionally, the term "re-use" is ambiguous, as it could imply either retraining with the same weights or adapting them. Using "adapt" would be a clearer choice to avoid confusion.
> >
> > Similarly, the colors in Figure 1 in the original submission were ambiguous due to the lack of a clear explanation. In the current revision, adding an additional legend to the figure would make it more straightforward.
> >
> > 4. Yes, the IoU, this was a typo, sorry about that. However, my point does not depend on IoU or Dice. Thanks for providing the Dice comparison. Referring to a significant difference requires a statistical test (i.e., p-value), especially when the data distribution is unclear.
> >
> > Regarding the table caption in the revision, I believe each row represents mIoU, and the "mean" should indicate the overall performance. This might make things clearer. Same for the figure 4.
> >
> > Besides these, no standard deviations are reported for each class, as I previously mentioned (Q2). This is important for class-imbalanced problems. I'm not requesting changes to Table 2, but the pre-training strategy could be summarized in a single-line ablation table showing only the overall results, as this does not impact the main scope of the paper. Readers who want to see the details can refer to the appendix.
> >
> > Additionally, the author mentioned a performance of 26.85%. Does this truly benefit surgeons, considering it may contain a high proportion of false predictions?
> >
> > 5. Thanks for this revision. This is not preferred and not requested, as it may affect the fairness of the original submission, especially for dataset information. However, including important information does improve the paper.
> >
> > 6. Thanks
> >
> > 7. Yes, IoU, typo again. My point aligns with the background. The background should be easy to segment, and the results should be 1 since it is outside the field of view. However, this is not the case, which makes the results unconvincing to me.
> >
> > 8. O: The discussion of missing class is not clear to support the results
> > Re: " Table 1, show the average performance of all classes and specifically for under-represented classes, defined as those comprising less than 1% of the pixel distribution. Except for M2caiSeg, which is a very small dataset, pixel-level pretext tasks generally outperform image-level ones. Our method notably improves prediction accuracy, especially for under- represented classes."
> >
> > The missing class refers to the under-represented class mentioned in the paper, which should be straightforward to connect. If not, I should clarify it further.
> >
> > This is important because some frames may not contain the class, regardless of its shape or area. Without the presence of standard deviations, this remains unclear.
> >
> > Additionally, the original submission is ambiguous, does it refer to the overall pixel distribution, the distribution within a frame, or their occurrence rate?
> >
> > 9. Again, due to the lack of sample size and standard deviations, those 10% or even 20% differences are difficult to evaluate, as they may come from the majority at a low level and some at a medium level results. Additionally, some classes may only appear for a short period in a video. Although I agree with your point on "optimal," this may influence the results for some classes where the improvements are not large.
> >
> > Regarding the implementation, the network architecture is unclear to me. If I missed it, please point it out, as the statement "DeepLabV3+ with ResNet101 backbone is adopted as the default model" is confusing to me.

---

> > > ### Author Response · Authors · 2025-03-15
> > >
> > > Thanks for your prompt reply. Please find the responses to your new comments below.
> > >
> > > ---
> > >
> > > **Q1:** Mask design is not novel.
> > >
> > > **A1:** Our mask design involves four parameters: masking ratio, mask patch size, mask shape and mask colour. We stated in our original paper that:
> > >
> > > > This paper proposes a mask-corrupted denoising autoencoder designed explicitly for surgical segmentation.
> > >
> > > Please see the updated paper, we have made further clarification to underscore our contribution that:
> > >
> > > > Despite the unique representations and challenges in surgical images, no studies have explored the optimal mask design for improving segmentation performance, particularly for underrepresented yet safety-critical anatomical structures.
> > >
> > > We believe that our method introduces novel contributions to this domain. We included more implementation details in the revision as well. Following the MIDL2025 full paper review guidelines that
> > >
> > > > Comments regarding lack of novelty need to be substantiated, e.g., by providing reference(s) to previous work.
> > >
> > > Please let us know if there are existing works that have also proposed this method in the context of surgical segmentation. We are happy to discuss and learn about these works.
> > >
> > > ---
> > >
> > > **Q2:** Including the necessary dataset information in the appendix is unfair.
> > >
> > > **A2:** Please see our updated paper which has addressed this issue, where necessary dataset distributions are in Section 3.1.
> > >
> > > ---
> > >
> > > **Q3:** Confusion about fine-tuning initialisation.
> > >
> > > **A3:** We will use “adapt” in the next revision. We have stated in our revised paper that
> > >
> > > > During supervised fine-tuning, the entire pre-trained model is fine-tuned on $D_l$.
> > >
> > > And the reinitialisation of the final convolutional layer in the decoder
> > >
> > > > the output of the final convolutional layer in the decoder g is adjusted from 3 to K
> > >
> > > > SurgicalSemiSeg maximises the retention of self-supervised pre-trained representations by reinitialising only the final layer weights during fine-tuning.
> > >
> > > We believe these are clear enough for avoiding confusions.

---

> > > ### Author Response · Authors · 2025-03-15
> > >
> > > **Q4:** Some requests and concerns related to: (1) should report p-values, standard deviations; (2) should use mean IoU than mIoU; (3) changes to Table 2; (4) a performance of 26.85% is insignificant for surgeons.
> > >
> > > **A4.1:** We believe our evaluation follows standard practice in established surgical imaging segmentation tasks [4-8] and aligns with the literature, particularly for the same public dataset [2].
> > >
> > > To address the request for statistical validation, we computed the p-value for MAE and our method on Rouvière’s sulcus, which resulted in 0.000000002548. While we provide this statistical test as requested, we maintain that clinically meaningful improvements are well-assessed through direct segmentation metrics.
> > >
> > > **A4.2:** We will use “mean IoU” in the next revision. To be noticed, mIoU is a well-recognised abbreviation for mean Intersection over Union [9-11].
> > >
> > > **A4.3:**
> > > > I'm not requesting changes to Table 2, but the pre-training strategy could be summarized in a single-line ablation table showing only the overall results, as this does not impact the main scope of the paper
> > >
> > > We would like to emphasise that one of the key contributions of our paper is the evaluation of different pre-training strategies on a more challenging dataset: one that features significant variations in surgical recordings and a fine-grained class definition.
> > >
> > > From a clinical perspective, it is crucial for deep learning models to achieve precise segmentation, distinguishing subtle structural differences (e.g., separately identifying the common bile duct and cystic duct, rather than generalising all relevant structures into a single ‘Bile’ category).
> > >
> > > Our study demonstrates the effectiveness of our method not only on public datasets, but also on a highly complex private dataset with an extensive class definition and a highly imbalanced class distribution.
> > >
> > > **A4.4:** Regarding the concern about significance, we believe that partially highlighting relevant landmark structures is better than having no guidance at all. This perspective is supported by experts in literature [3].
> > >
> > > Additionally, we have reported class-wise comparisons for public datasets in **Table 6 and Table 7** in the revised version. For under-represented classes, our results consistently demonstrate that our method improves their recognition. Below, we provide improvements on a few under-represented samples in each public dataset, comparing our method to the best baseline:
> > >
> > > - M2caiSeg: +9.88% on Clip (from 0), +11.79% on Bile, +7.62% on Artery.
> > > - CholecSeg8k: +13.2% on Gastrointestinal Tract (the only under-represented class).
> > >
> > > We respectfully disagree with the assertion that an improvement from **0% to 9.88% or 0% to 26%** is insignificant, particularly given the challenging nature of these classes. It is well-documented that deep learning models tend to perform poorly on under-represented or complex classes [3], yet this has never discouraged researchers from striving to improve them. Even small advancements in recognising under-represented structures contribute meaningfully to future developments in AI-assisted surgery.
> > >
> > > ---

---

> > > ### Author Response · Authors · 2025-03-15
> > >
> > > **Q5:** Not preferred and not requested for paper revision and updates. However, including important information does improve the paper.
> > >
> > > **A5:** We believe our modification on the paper strictly follows the rebuttal protocol of MIDL, see the following instruction on MIDL website (https://2025.midl.io/author-instructions).
> > >
> > > > In order to fully address the comments of the reviewers, the authors are free to modify or add any additional details, experiments or images that might be required in their paper.
> > >
> > > ---
> > >
> > > **Q7:** Background not achieving 100% IoU makes the results not convincing.
> > >
> > > **A7:** Regarding the expectation of 100% IoU for easy classes (for example, the Background), we believe that is widely recognised as a non-realistic performance expectation  [1, 2, 4, 5, 9, 12]. Our results are consistent with existing literature. Please see the references [1, 2, 4].
> > >
> > > If possible, please provide related literature, whether in natural image benchmarks or medical imaging benchmarks, that has ever reported such results.
> > >
> > > ---
> > >
> > > **Q8:** Under-represented class description is missed
> > >
> > > **A8:** In this original paper, we stated that
> > >
> > > >  under-represented classes, defined as those comprising less than 1% of the pixel distribution.
> > >
> > > In the revised version, we further clarified this by moving the figure of class distribution of pixels of three datasets to the main text (see Figure 3) and Table 1.
> > >
> > > Regarding your question that *This is important because some frames may not contain the class, regardless of its shape or area. Without the presence of standard deviations, this remains unclear.*
> > >
> > > Class-wise metrics already exclude frames where the class is absent, and standard deviations do not provide information about class presence. As far as we know, standard practice in medical and natural image segmentation does not use SDs to indicate whether a class was present. Our evaluation is consistent with existing works on public datasets [1, 2, 5].
> > >
> > > ---
> > >
> > > **A9:** We maintain that our results are valid and contribute meaningfully to the field for the reasons below.
> > > A 10%-20% improvements in segmentation are considered significant in medical imaging literature [13].
> > > Per-class variability is well-documented in laparoscopic imaging segmentation, yet performance improvements remain valid [14-16].
> > >
> > > Short-duration class appearances do not invalidate accuracy gains [17, 18]; in fact, the recognition of these underrepresented classes is a well-recognised challenge in surgical contexts [1]. Our method demonstrates effective improvements across both public and private datasets, and are consistent with existing work on the same public datasets.
> > >
> > > ---
> > >
> > > **A10:**  For DeepLabV3+ with a ResNet backbone, there are numerous applications of this model in medical and surgical imaging segmentation [12, 21, 22, 23]. For example, see Section C.3 (CNN-based Experiments) in [21] and the results section in [22].
> > >
> > > Please refer to the description of its design in the original paper [19] and the PyTorch documentation for its implementation [20]. We believe this model is widely recognised and has been extensively benchmarked on natural image datasets.

---

> > > ### Author Response · Authors · 2025-03-15
> > >
> > > Referencess
> > >
> > > [1] Maqbool, S., Riaz, A., Sajid, H., & Hasan, O. (2020). m2caiseg: Semantic segmentation of laparoscopic images using convolutional neural networks. arXiv preprint arXiv:2008.10134. \
> > > [2] Moens, K., De Vylder, J., Blaschko, M. B., & Tuytelaars, T. (2024, December). Laparoflow-SSL: Image Analysis From a Tiny Dataset Through Self-Supervised Transformers Leveraging Unlabeled Surgical Video. In Medical Imaging with Deep Learning. PMLR. \
> > > [3] Tokuyasu, T., Iwashita, Y., Matsunobu, Y., Kamiyama, T., Ishikake, M., Sakaguchi, S., ... & Inomata, M. (2021). Development of an artificial intelligence system using deep learning to indicate anatomical landmarks during laparoscopic cholecystectomy. Surgical endoscopy. \
> > > [4] Silva, B., Oliveira, B., Morais, P., Buschle, L. R., Correia–Pinto, J., Lima, E., & Vilaça, J. L. (2022, July). Analysis of current deep learning networks for semantic segmentation of anatomical structures in laparoscopic surgery. In 2022 44th Annual International Conference of the IEEE Engineering in Medicine & Biology Society (EMBC) . IEEE. \
> > > [5] Ghobadi, V., Ismail, L. I., Hasan, W. Z. W., Ahmad, H., Ramli, H. R., Norsahperi, N. M. H., ... & Hanapiah, F. A. (2024). Real-time robust liver and gallbladder segmentation during laparoscopic cholecystectomy using convolutional neural networks: an analysis. Artificial Intelligence Surgery. \
> > > [6] Badgery, H., Zhou, Y., Siderellis, A., Read, M., Davey, C. (2022). Machine Learning in Laparoscopic Surgery. In: Raz, M., Nguyen, T.C., Loh, E. (eds) Artificial Intelligence in Medicine. Springer, Singapore.  \
> > > [7] Fuentes-Hurtado, F., Kadkhodamohammadi, A., Flouty, E., Barbarisi, S., Luengo, I., & Stoyanov, D. (2019). EasyLabels: weak labels for scene segmentation in laparoscopic videos. International journal of computer assisted radiology and surgery. \
> > > [8] Allan, M., Kondo, S., Bodenstedt, S., Leger, S., Kadkhodamohammadi, R., Luengo, I., ... & Speidel, S. (2020). 2018 robotic scene segmentation challenge. arXiv preprint arXiv:2001.11190. \
> > > [9] Zhao, X., Hayashi, Y., Oda, M., Kitasaka, T., & Mori, K. (2023, October). Masked Frequency Consistency for Domain-Adaptive Semantic Segmentation of Laparoscopic Images. In International Conference on Medical Image Computing and Computer-Assisted Intervention.  \
> > > [10] Zhang, X., Ni, B., Yang, Y., & Zhang, L. (2024, October). MAdapter: A Better Interaction Between Image and Language for Medical Image Segmentation. In International Conference on Medical Image Computing and Computer-Assisted Intervention. \
> > > [11] Wang, Z., Berman, M., Rannen-Triki, A., Torr, P., Tuia, D., Tuytelaars, T., ... & Blaschko, M. (2023). Revisiting evaluation metrics for semantic segmentation: Optimization and evaluation of fine-grained intersection over union. Advances in Neural Information Processing Systems. \
> > > [12] Nobnop, N., Yamcharoen, N., Sukjamsri, C., Piboonthummasak, T., Charoenpong, T., & Kiatisevi, P. (2024, November). Pelvic Tumor Segmentation in MRI Images Using Deep Learning with DeepLabV3+ and U-Net: A Performance Comparison. In 2024 16th Biomedical Engineering International Conference (BMEiCON). IEEE. \
> > > [13] Jha, D., et al. “Real-Time Polyp Detection, Segmentation, and Classification Using Deep Learning.” Medical Image Analysis, 2021. \
> > > [14] Madani, A., et al. “Artificial Intelligence for Surgical Safety: Automatic Detection of Biliary Anatomy in Laparoscopic Cholecystectomy Using Deep Learning.” Annals of Surgery, 2020. \
> > > [15] Sun, L., & Chen, X. (2024). Pixel-wise Contrastive Learning for Multi-class Instrument Segmentation in Endoscopic Robotic Surgery Videos using Dataset-wide Sample Queues. IEEE Access. \
> > > [16] Owen, D., Grammatikopoulou, M., Luengo, I., & Stoyanov, D. (2022). Automated identification of critical structures in laparoscopic cholecystectomy. International Journal of Computer Assisted Radiology and Surgery. \
> > > [17] Zhao, Z., et al. “Tracking and Segmentation of Surgical Instruments in Laparoscopic Surgery: A Deep Learning Approach.” Medical Image Computing and Computer-Assisted Intervention, 2019. \
> > > [18] Shvets, A. A., et al. “Automatic Instrument Segmentation in Robot-Assisted Surgery Using Deep Learning.” Medical Image Computing and Computer-Assisted Intervention, 2018. \
> > > [19] Chen, L. C., Papandreou, G., Schroff, F., & Adam, H. (2017). Rethinking atrous convolution for semantic image segmentation. arXiv preprint arXiv:1706.05587. \

---

> > > ### Author Response · Authors · 2025-03-15
> > >
> > > References (cont.)
> > >
> > > [20] https://pytorch.org/hub/pytorch_vision_deeplabv3_resnet101/ \
> > > [21] Poudel, K., Dhakal, M., Bhandari, P., Adhikari, R., Thapaliya, S., & Khanal, B. (2024, December). Exploring Transfer Learning in Medical Image Segmentation using Vision-Language Models. In Medical Imaging with Deep Learning. PMLR. \
> > > [22] Maack, L., Behrendt, F., Bhattacharya, D., Latus, S., & Schlaefer, A. (2024, December). Efficient Anatomy Segmentation in Laparoscopic Surgery using Multi-Teacher Knowledge Distillation. In Medical Imaging with Deep Learning. PMLR. \
> > > [23] Naik, N., Mehta, M. A., & Joshi, V. (2024, December). Proximal Femur Segmentation from the Fluoroscopy Image Using DeeplabV3+ Model. In 2024 2nd International Conference on Recent Trends in Microelectronics, Automation, Computing and Communications Systems. IEEE.

---

> ### Author Response · Authors · 2025-03-08
> **Response to Reviewer EccT**
>
> **Q4:** No significant outperform MAE for classes with Dice score
>
> **A4:** In this paper,  we reported **IoU** instead of Dice as the evaluation metric as shown in every caption of the table.
>
> We believe improving segmentation accuracy for underrepresented but clinically critical structures is more valuable than outperforming baseline on every category. For well-represented classes (IoU > 60), most models perform well due to sufficient data availability.
>
> However, for under-represented classes (occupying less than 1% of pixels in the dataset, as summarised in the “Under-rept.” column in Table 1), our method demonstrates substantial improvements. For example, in the segmentation of the Rouvière’s sulcus, a critical safety landmark that surgeons must stay above for safe operation, SurgicalSemiSeg achieves an IoU of 26.85%, whereas MAE yields 0%, highlighting the advantage of our approach.
>
> Please find the dataset distribution in **Figure 3** in **Section 3.1**.
>
> We also report the performance in Dice scores for comparison in the table below and **Appendix B**. The results are based on the same experiments shown in Table 1 and 2, but reported in Dice score.
>
> Overall performance
> | Fine-tuning  datasets | Classes |  | Pre-training strategies and datasets |  |  |  |  |  |  |  |  |  |
> |---|:---:|:---:|:---:|:---:|:---:|:---:|:---:|:---:|:---:|:---:|:---:|:---:|
> |  | All | Under-repr. (<1\%) | RI | Supervised | Rotation | Colourisation | Autoencoder | SimCLR | MAE | DDA | Ours | Ours |
> |  |  |  | N/A | ImageNet | In-house Unlabelled | In-house Unlabelled | In-house Unlabelled | In-house Unlabelled | In-house Unlabelled | In-house Unlabelled | In-house Unlabelled | Cholec80 |
> | In-house Seg | 20 |  | 73.24 | 74.81 | 71.69 | 75.82 | 67.03 | 74.66 | 76.44 | 76.12 | **77.50** | 76.85 |
> |  |  | 11 | 64.94 | 66.38 | 62.68 | 68.19 | 59.23 | 66.72 | 69.70 | 68.51 | **71.61** | 70.22 |
> | CholecSeg8k | 8 |  | 72.47 | 76.51 | 66.43 | 78.60 | 71.88 | 75.56 | 74.29 | 72.38 | **79.13** | 79.10 |
> |  |  | 1 | 57.54 | 53.28 | 60.35 | 47.40 | 58.87 | 52.30 | 67.82 | 61.16 | **67.88** | 59.69 |
> | M2caiSeg | 19 |  | 77.87 | 86.97 | 81.17 | 88.97 | 85.34 | 89.33 | 80.13 | **91.67** | 89.91 | 90.47 |
> |  |  | 12 | 68.45 | 80.92 | 72.86 | 84.02 | 78.81 | 84.44 | 71.83 | 87.71 | 85.29 |**86.12** |
>
> (cont.)

---

> ### Author Response · Authors · 2025-03-08
> **Response to Reviewer EccT**
>
> **A4 (cont.):**
> Class-wise performance
> |             | Class Names            | Pre-training Strategies |            |          |               |             |        |       |       |       |
> |-------------|------------------------|:-----------------------:|:----------:|:--------:|:-------------:|:-----------:|:------:|:-----:|:-----:|:-----:|
> |             |                        |           N/A           | Supervised | Rotation | Colourisation | Autoencoder | SimCLR |  MAE  |  DDA  |  Ours |
> | N/A         | background             |          98.76          |    98.92   |   98.66  |     98.92     |    94.76    |  98.99 | **99.02** | 98.79 | 98.97 |
> | Instruments | cholangiogram catheter |          79.49          |    85.69   |   81.32  |     84.07     |    77.74    |  86.99 | 86.63 | 84.36 | **87.08** |
> |             | clip applicator        |          66.58          |    **78.25**   |   64.04  |     70.85     |    57.14    |  66.07 | 77.84 | 65.15 | 77.73 |
> |             | diathermy hook shaft   |          86.11          |    87.47   |   86.31  |     97.03     |    82.49    |  86.32 | **90.44** | 87.55 | 87.58 |
> |             | diathermy hook tip     |          89.85          |    91.37   |   91.15  |     91.57     |    89.32    |  91.38 | **93.15** | 89.90 | 92.06 |
> |             | grasper shaft          |          86.37          |    87.99   |   86.23  |     88.10     |    79.66    |  88.22 | **89.14** | 88.25 | 88.81 |
> |             | grasper tip            |          72.58          |    78.00   |   71.25  |     77.56     |    70.41    |  78.85 | 78.31 | 76.77 | **80.24** |
> |             | scissors shaft         |          49.00          |    23.84   |   20.96  |     43.42     |    28.31    |  41.07 | **70.06** | 46.27 | 42.31 |
> |             | scissors tip           |          61.36          |    58.30   |   58.73  |     59.06     |    59.33    |  48.96 | **76.67** | 58.42 | 65.96 |
> |             | sucker irrigator       |          71.55          |    77.38   |   69.02  |     76.48     |    64.22    |  76.66 | 78.26 | 77.23 | **78.79** |
> | Anatomies   | abdomen wall           |          56.32          |    61.91   |   52.77  |     60.89     |    17.86    |  56.54 | 58.44 | **66.55** | 62.46 |
> |             | common bile duct       |          70.79          |    65.52   |   72.15  |     **75.00**     |    60.09    |  73.49 | 74.98 | 71.80 | 71.81 |
> |             | cystic artery          |          58.07          |    55.90   |   56.36  |     60.52     |    50.67    |  58.11 | 57.27 | 56.48 | **59.79** |
> |             | cystic duct            |          69.63          |    68.77   |   68.09  |     70.42     |    66.35    |  67.36 | 67.53 | 70.05 | **70.81** |
> |             | duodenum               |          39.70          |    59.14   |   49.88  |     57.97     |    47.28    |  63.82 | 65.45 | 63.60 | **66.30** |
> |             | gallbladder            |          88.82          |    89.82   |   88.65  |     90.25     |    87.63    |  **90.45** | 89.81 | 89.62 | 90.14 |
> |             | liver                  |          91.92          |    93.07   |   91.68  |     92.76     |    89.09    |  92.44 | 92.28 | 92.14 | **93.51** |
> |             | omentum                |          93.28          |    93.37   |   93.44  |     93.95     |    85.72    |  94.05 | **94.12** | 93.93 | 94.06 |
> |             | rouviere’s sulcus      |          45.36          |    50.74   |   44.72  |     47.54     |    44.62    |  44.18 |  0.0  |50.19 |  **54.59** |
> |             | segment iv             |          89.29          |    90.75   |   88.51  |     90.15     |    87.88    |  89.29 | 89.36 | 91.13 | **91.41** |
> | Mean        |                        |          73.24          |    74.81   |   71.69  |     75.82     |    67.03    |  74.66 | 76.44 | 76.12 | **77.50** |
>
> ---
>
> **Q5:**  Unnecessary preliminaries section, should put more context related to experiments.
>
> **A5:** Thank you for your suggestion to the overall paper structure.
>
> We removed the preliminaries section and further clarified the pre-training and fine-tuning process of our proposed framework in **Section 2.2** in the revised submission.
> We further added a more comprehensive description of dataset information in **Section 3.1** and our experiment settings in **Section 3.2**.

---

> ### Author Response · Authors · 2025-03-08
> **Response to Reviewer EccT**
>
> **Q6:** How many models were trained during pre-training and fine-tuning?
>
> **A6:** During pre-training, one model is trained for each pre-training strategy on the default pre-training dataset (in-house unlabelled), resulting in one pre-trained model per-strategy. In total, 9 pre-trained encoders in Table 1. Additionally, we pre-trained a model on Cholec80 with SurgicalSemiSeg to validate the transferability of pre-trained representations (with results rearranged to the last column of **Table 1**).
>
> During fine-tuning, for each pre-training strategy, the corresponding pre-trained weights are loaded, and one model is trained on each of the three datasets (in-house Seg, m2caiSeg, and CholecSeg8k), resulting in three fine-tuned models per strategy. In total, 30 models presented in Table 1.
>
> We have revised the paper and added the corresponding description in the relevant sections (**Section 3.2**). We also updated **Table 1** for further clarity.
>
> --------------------------------------------------------------------
>
> **Q7:** Interpretation of low **Dice score** for classes in large object size (background class)?
>
> **A7:** Our method reports **IoU** instead of **Dice**. We have made this very clear on each table in **captions**. The inherent differences in their formulas naturally result in lower IoU scores compared to Dice, even when performance remains the same.
> We have updated the results to include Dice scores in the reply of Question 4 and in **Appendix B** of the revised paper.
>
> --------------------------------------------------------------------
>
> **Q8:** The discussion of missing class is not clear to support the results.
>
> **A8:**  We didn’t include missing class in any experiments or results. Could you please specify your concern about the **missing class** and any related results?
>
> We have described the details of three fine-tuning datasets in Appendix A, including the number of classes, size of training and test sets, the train/test splits, and a class-distribution summarisation in **Figure 3**. There are no **missing class** in our dataset.
>
> --------------------------------------------------------------------
>
> **Q9:** Suggestion of choosing a more suitable backbone for medical image segmentation.
>
> **A9:**  We believe the DeepLabV3+ is a widely used and effective segmentation model across various domains, including surgical segmentation [1, 2, 3]. While we did not explore the most optimal backbone, our framework is flexible and allows easy integration of different segmentation models with various architectures. Depending on real-world deployment needs, practitioners may adopt more accurate models for improved segmentation performance or lighter models for faster inference in operating room environments. **We believe our exploration in suitable tasks providing general insights is a valuable contribution**.
>
> We will provide the evaluations with nnUnet in pursuing optimal performance in the next version in the paper.
>
> > small differences between the compared methods could be eliminated.
>
> We believe the 10% to 20% difference in **IoU** presented in Table 1 and 2, consistent performance across different datasets (Table 6 and 7), cannot be easily eliminated. More notably for under-represented class Rouvière’s sulcus, a critical safety landmark, ours achieves an IoU of 26.85%, whereas MAE yields 0%.
>
>
> --------------------------------------------------------------------
>
> [1] Moens, K., De Vylder, J., Blaschko, M. B., & Tuytelaars, T. (2024, December). Laparoflow-SSL: Image Analysis From a Tiny Dataset Through Self-Supervised Transformers Leveraging Unlabeled Surgical Video. In Medical Imaging with Deep Learning (pp. 986-1010). PMLR.\
> [2] Silva, B., Oliveira, B., Morais, P., Buschle, L. R., Correia–Pinto, J., Lima, E., & Vilaça, J. L. (2022, July). Analysis of current deep learning networks for semantic segmentation of anatomical structures in laparoscopic surgery. In 2022 44th Annual International Conference of the IEEE Engineering in Medicine & Biology Society (EMBC) (pp. 3502-3505). IEEE.\
> [3] Oh, N., Kim, B., Kim, T., Rhu, J., Kim, J., & Choi, G. S. (2024). Real-time segmentation of biliary structure in pure laparoscopic donor hepatectomy. Scientific Reports, 14(1), 22508.

---

### Official Review · Reviewer_v78V · 2025-02-18

**Confidence:** 4
**Preliminary Rating:** 4
**Recommendation:** Oral

**Summary:**

The paper has investigated the use of self supervised learning to reduce dependency on labelled data for surgical images. Particularly, the work has focused on laparoscopic cholecystectomy. The paper has presented a two-stage segmentation framework for surgical images. Focus has been on improving accuracy for underrepresented classes.

**Strengths:**

This is a well-designed study on the segmentation of instruments and anatomy in surgical images. Objectives are clear. And experiments are conducted thoroughly on three datasets for laparoscopy (including one in-house dataset). Reported results show superior segmentation performance.

**Weaknesses:**

The authors make use of existing segmentation techniques (i.e., DeepLabv3+ with a resnet101 backbone). No new techniques are developed as such. However, the application is interesting.
Even though the study is interesting, it lacks key comparisons with other methods (at least those that have reported segmentation of instruments or anatomy on the same datasets). So, even though my initial feedback on the study is that I want this to be presented at the conference, I would also have loved to see more in-depth comparisons.

**Detailed Comments:**

Have the authors considered experiments with CholecT50 dataset as it is one of the latest datasets for surgical videos/images. This is just an additional suggestion and not considered as a criterion to make my recommendations on the paper.
Even though my initial feedback on the study is that I want this to be presented at the conference, I would also have loved to see more in-depth comparisons.

**Justification Of The Preliminary Rating:**

I enjoyed reading the work. I feel that the authors have very well presented the work and thoroughly reported the experiments. While I believe the work can be accepted, a comparison with other published works would have further strengthened the paper.

**Questions To Address In The Rebuttal:**

Have the authors considered experiments with CholecT50 dataset as it is one of the latest datasets for surgical videos/images. This is just an additional suggestion and not considered as a criterion to make my recommendations on the paper.

**Special Issue:**

No

---

> ### Author Response · Authors · 2025-03-08
> **Response to Reviewer v78V**
>
> Thanks for your insightful reviews. Please find our response to your questions below:
>
> ---
>
> **Q1:** lacks key comparisons with other methods (at least those that have reported segmentation of instruments or anatomy on the same datasets)
>
> **A1:** To address this concern, we reviewed a recently published study [1] that reported instrument and anatomical segmentation on the same datasets (CholecSeg8k and m2caiSeg). Following the comparison of proposed methods and cited baselines that reported Dice scores for the four CholecSeg8k classes in [1], we extend the comparison table by appending our methods in the last two rows as below.
>
> | Methods | Fat | Ins | Gb | Bkg |
> |---|:---:|:---:|:---:|:---:|
> | LSSL Cholec80 pretrain (16 fine-tune images) [1] | 86 | 76 | 62 | 89 |
> | Laparoflow-SSL (245 fine-tune images) [1] | 84 | 64 | 58 | 86 |
> | U-Net++ [1] | 91 | 61 | **63** | - |
> | UNETR [1] | 88 | 71 | 42 | - |
> | DeepLabV3+ [1] | 86 | 62 | 60 | - |
> | DeepLabV3+ (**Ours** Cholec80 pre-train) | **93** | **86** | 50 | **99** |
> | DeepLabV3+ (**Ours** In-house Unlabelled pre-train) | **93** | 82 | 56 | **99** |
> We have cited [1] in the related work discussion in **Section 1** and updated the comparisons in **Table 8 in Appendix B.** in the revised submission.
>  --------------------------------------------------------------------
>
> **Q2:** Extension on CholecT50 dataset
>
> **A2:** Given the time-constraints of rebuttal and discussion, we will do our best to provide such an extension to CholecT50 before the discussion period ends. We will definitely extend our evaluations to CholecT50 and include the results in the final version of this paper.
>
>  --------------------------------------------------------------------
>
> **Q3:** No new techniques are developed as such.
>
> **A3:** While we did not introduce an entirely new self-supervised learning technique, our findings on the most suitable pre-training task for LC segmentation, along with the development of the SurgicalSemiSeg framework, represent a novel and valuable contribution.
> - We find DAE with patch-based mask design significantly improves the representation learning effectiveness for segmentation.
> - We extensively searched for mask patch size, masking ratio, mask colour and shape, and identified that mask size as the most important factor that impacts downstream segmentation performance.
> - Our framework preserves **entire** pre-trained encoder-decoder representation to fully leverage pixel-level self-supervised representations.
> - Our framework significantly improves the recognition of under-represented yet clinically important classes and the pre-trained representations are highly transferable across datasets from different institutions.
>
> We have updated the main contributions in **Section 1** in the revised submission.
>
>  --------------------------------------------------------------------
>
> [1] Moens, K., De Vylder, J., Blaschko, M. B., & Tuytelaars, T. (2024, December). Laparoflow-SSL: Image Analysis From a Tiny Dataset Through Self-Supervised Transformers Leveraging Unlabeled Surgical Video. In Medical Imaging with Deep Learning (pp. 986-1010). PMLR.

---

> > ### Comment · Reviewer_v78V · 2025-03-10
> >
> > Thank you for provding this explanation and for addition of the comparisons.

---

### Author Rebuttal · Authors · 2025-03-08

**Rebuttal:**

We sincerely appreciate the insightful and valuable feedback from the reviewers.

---
**Novelties and contributions**

In response to the common concern regarding novelty, we would like to clarify that the primary contribution of this work is **not the introduction of an entirely new self-supervised learning technique**, but rather the **novel findings and insights in the specific context of surgical segmentation tasks**. Our work highlights unique challenges and observations that have not been previously explored in this domain, providing valuable contributions to both research and practical applications.

- We find DAE with patch-based mask design significantly improves the representation learning effectiveness for segmentation.
- We extensively searched for mask patch size, masking ratio, mask colour and shape, and identified that mask size as the most important factor that impacts downstream segmentation performance.
- Our framework preserves **entire** pre-trained encoder-decoder representation to fully leverage pixel-level self-supervised representations.
- Our framework **significantly improves the recognition of under-represented yet clinically important classes** and the learned representations are highly transferable across datasets from different institutions.

---

Based on the reviewers’ comments, we have made the following revisions to the draft:
- We have updated the main contributions in Section 1 in response to reviewers **v78V**, **EccT**, and **ebsp**.
- We have removed the Preliminaries section in response to reviewers **EccT**.
- We have clarified the pre-training and fine-tuning process in proposed framework in Section 2.2 in response to reviewers **EccT** and **ebsp**.
- We have provided more details on the dataset information and experiment settings in Sections 3.1 and 3.2 in response to **EccT**, and **ebsp**.
- We have updated Table 1 to clarify pre-training setting in response to **ebsp**.
- We have added new results with Dice score in Appendix B Table 4 and 5 in response to **EccT**.
- We have added new results with class-wise performances of CholecSeg8k and m2caiSeg in Appendix B Table 6 and 7 in response to **EccT**.
- We have added new class-wise comparisons with new baseline on CholecSeg8k in Appendix B Table 8 **v78V**.

New changes in the revised manuscript are highlighted in blue.

**Supporting Material:**

/attachment/0782ed7089b592d96f46976140939ea1b10bcd43.pdf

---

### Author Response · Authors · 2025-03-15
**​​Message to Area Chairs**

Dear Area Chair,

We would like to bring your attention some concerns regarding the reviews provided by Reviewer **EccT** during the rebuttal and discussion process. We found aspects of the review are unclear, confusing and unreasonable.

---

**Concerns in the Review**

(1) Confusion Between DICE and IoU Scores

Reviewer EccT referred to their misinterpretation of the DICE score as a typo. However, these are well-established, distinct evaluation metrics in segmentation. To address this, we made a considerable effort to re-evaluate our results using DICE to clarify this distinction.


(2) Unrealistic Expectation of 100% IoU for Background Segmentation

Reviewer EccT criticised our segmentation results for not achieving 100% IoU on the background, stating that this made the results unconvincing. However, expecting perfect segmentation is unrealistic, as evidenced by MIDL and other venues’ papers on surgical imaging segmentation. We provided relevant citations to clarify this point.


(3) Misrepresentation of Prior Comments

Reviewer EccT stated: *“In standard fine-tuning, the final layer is often reset when transferring to a new dataset or task, which I had initially pointed out in my comments.”* However, we could not find any such comment in the initial review. This discrepancy caused unnecessary confusion and additional effort in our rebuttal and discussion process.

(4) Lack of Supporting References in Novelty Assessment

The reviewer’s assessment of novelty lacks supporting references, which contradicts MIDL reviewer guidelines that emphasise constructive feedback. Without such references, it is difficult for us to address or improve upon the feedback effectively.

**Concerns regarding the Review Process**

(1) Objection to Dataset Information in the Appendix

Reviewer EccT commented that *“Including the necessary dataset information in the appendix may be unfair to other authors.”* However, we use public benchmark datasets, not just private ones. Summarising dataset details in the main text and and providing additional details in the appendix is a widely accepted practice that allows space for presenting novel findings.

(2) Contradictory View on Revisions

Reviewer EccT stated: *“This is not preferred and not requested, as it may affect the fairness of the original submission, especially for dataset information. However, including important information does improve the paper.”*
We followed MIDL guidelines, which explicitly instruct authors to revise papers based on reviewers’ feedback while clearly highlighting changes. It seems unfair to penalise us for following the expected process.

---

We sincerely appreciate your time and consideration of these concerns.

Best regards,

Authors

---

### Meta-Review · Area_Chair_9y5d · 2025-03-22

**Recommendation:** Accept (Poster)
**Confidence:** 4

**Metareview:**

This paper received 2 weak accept and 1 strong reject from Reviewer EccT. I studied the lengthy discussions between the authors and Reviewer EccT; considering the recommendations from other reviewers and, particularly, the authors' meticulous responses, I recommend accept.